# A Review of Ethnoveterinary Knowledge, Biological Activities and Secondary Metabolites of Medicinal Woody Plants Used for Managing Animal Health in South Africa

**DOI:** 10.3390/vetsci8100228

**Published:** 2021-10-12

**Authors:** Kelebogile Martha Selogatwe, John Awungnjia Asong, Madeleen Struwig, Rendani Victress Ndou, Adeyemi Oladapo Aremu

**Affiliations:** 1Food Security and Safety Niche Area, Faculty of Natural and Agricultural Sciences, North-West University, Private Bag X2046, Mmabatho 2790, South Africa; kselogatwe@gmail.com; 2Unit for Environmental Sciences and Management, Faculty of Natural and Agricultural Sciences, North-West University, Private Bag X2046, Mmabatho 2790, South Africa; johnmilan058@gmail.com (J.A.A.); 12516309@nwu.ac.za (M.S.); 3Indigenous Knowledge Systems Centre, Faculty of Natural and Agricultural Sciences, North-West University, Private Bag X2046, Mmabatho 2790, South Africa; 4Centre of Animal Health Studies, Faculty of Natural and Agricultural Sciences, North-West University, Private Bag X2046, Mmabatho 2790, South Africa; rendani.ndou@nwu.ac.za

**Keywords:** antibacterial, antioxidant, ethnoveterinary, Fabaceae, livestock diseases, retained placenta

## Abstract

Globally, the use of ethnoveterinary medicine as remedies for animal health among different ethnic groups justify the need for a systematic exploration to enhance their potential. In addition, the increasing popularity and utilisation of woody plants remain common in traditional medicine, which may be attributed to their inherent benefits. The current review was aimed at analysing ethnoveterinary surveys, biological activities, and secondary metabolites/phytochemical profiles of the woody plants of South Africa. Eligible literature (period: 2000 to 2020) were retrieved from different databases such as Google Scholar, PubMed, Sabinet, and Science Direct. Based on the inclusion and exclusion criteria, 20 ethnoveterinary surveys were eligible and were subjected to further analysis. We identified 104 woody plant species from 44 plant families that are used in the treatment of different diseases in animals, particularly cattle (70%) and goats (20%). The most mentioned (with six citations) woody plants were *Terminalia sericea* Burch. ex DC and *Ziziphus mucronata* Willd., which were followed by plants with five (*Cussonia spicata* Thunb., *Pterocarpus angolensis* DC and *Vachellia karroo* (Hayne) Banfi & Galasso) or four (*Acokanthera oppositifolia* (Lam.) Codd, *Cassia abbreviata* Oliv., and *Strychnos henningsii* Gilg) individual mentions. The most dominant families were Fabaceae (19%), Apocynaceae (5.8%), Rubiaceae (5.8%), Anacardiaceae (4.8%), Combretaceae (4.8%), Euphorbiaceae (4.8%), Malvaceae (4.8%), Rhamnaceae (4.8%), and Celastraceae (3.8%). Bark (33%), leaves (29%), and roots (19%) were the plant parts dominantly used to prepare remedies for ethnoveterinary medicine. An estimated 20% of woody plants have been screened for antimicrobial, anthelmintic, antioxidant, and cytotoxicity effects. Phytochemical profiles established a rich pool of valuable secondary metabolites (phenolic, flavonoids and condensed tannins) that may be responsible for the exerted biological activities. Overall, the significant portion of woody plants lacking empirical evidence on their biological effects indicates a major knowledge gap that requires more research efforts.

## 1. Introduction

Globally, many animals, especially cattle, goats, and horses, play diverse role in human life, ranging from being a source of food, income and cultural wealth, touristic attraction, and job creation [1,2,3,4,5,6]. The rearing of animals is well-embedded in the culture of many ethnic groups, which justifies the popularity of indigenous knowledge and practice for managing the health and well-being of animals [5,7,8,9,10,11]. Relative to ethnobiology/ethnobotany, this is currently considered as a distinct field known as ethnoveterinary medicine, a word coined by the American anthropologist Constance McCorkle [12]. It is defined as “the systematic study and application of folk knowledge and beliefs, practices that relate to any aspects of animal health” [13]. Based on increasing evidence [6,14], the field has the capacity to develop into a huge industry in the future. Although animal species and plant species are utilised in ethnoveterinary medicine, the latter is often more popular among many ethnic groups globally [5,11]. Particularly in South Africa, the importance of plants for the management of animal health and well-being cannot be over-emphasized [15].

South Africa has a huge flora diversity and is recognised as a mega-diverse country with three global biodiversity hotspots [16]. The country has an estimated 24,000 species distributed among 368 families, which accounts for approximately 10% of the world’s flora [17]. As a result, South Africans have tapped into the healing powers of these floras since time immemorial, and this knowledge has been retained and has continuously evolved through generations [18]. In some cases, the choice of plants is based on an indigenous experimental process, cultural beliefs, and the biodiversity in a particular area [19,20,21,22].

In Africa, woody plants (trees and shrubs) are an important defining feature of the landscapes [20,23]. They are widely recognised for their diverse uses by humans since ancient times [20,24,25]. Particularly in traditional medicine, woody plant species have essential roles that are easily exemplified, as they represent about 65% of the top 51 most important African medicinal plants [26]. Recent studies from different African countries including South African have reiterated the vital role of woody plants in human and animal healthcare as well as the need for more concerted efforts for their conservation [25,27,28,29,30]. An important attribute of woody plants is the wide range of their organs (leaves, bark, roots, fruits, and flowers) that is available for use as remedies in folk medicine [25,27]. Relative to herbaceous plants with short life cycles, woody plants are often dominant in ecosystems, thereby making them apparent to foraging animals and for utilisation by humans [20,21]. In the current review, we aim to provide an appraisal on the existing ethnoveterinary knowledge, biological activities, and secondary metabolites/phytochemical profiles of woody plants used for managing animal health in South Africa. The review is expected to identify existing research gap(s) in an attempt to explore the potential of woody plants as an alternative remedy for managing animal health.

## 2. Materials and Methods

The literature search strategy was facilitated using keywords such as woody plants, ethnoveterinary medicine, livestock, and animal health. In addition, phytochemical, antioxidant, phenolic, and antibacterial effect were examples of terms used to generate data for the biological activity and phytochemical aspects of this review. These keywords were used singly and in combination to identify suitable literature from several databases, namely Google Scholar, Pubmed, Science Direct, and Sabinet. We focused on peer-reviewed papers published from 2000 to 2020 on South African woody plants.

Screening of the research outputs from the databases was conducted in two stages. Firstly, the title and abstract of the papers were screened against the inclusion criteria. A publication needed to provide the Latin name for the woody plant to be eligible for inclusion. Articles reporting on ethnoveterinary uses, biological activities, and phytochemical analyses of the woody plants fulfilled the inclusion criteria (e.g., specified time duration, woody species, and South African studies). Review papers and studies not involving South African medicinal woody plants were excluded. Based on the selection criteria, 20 papers were selected and were analysed in order to generate an inventory of woody plants (Table 1). Subsequently, analyses on plant families, mode of preparation, plant parts used to treat animals/livestock diseases, biological activities, and phytochemicals were conducted. Based on the significance of scientific names [31,32], the botanical names were validated via multi-databases, such as PlantZAfrica (http://pza.sanbi.org/ (accessed on 29 September 2021)), The Plant List (http://www.theplantlist.org/ (accessed on 29 September 2021)), and The World Flora Online (http://www.worldfloraonline.org/ (accessed on 29 September 2021)).

## 3. Results and Discussion

### 3.1. Overview of Eligible Literature and Ethnoveterinary Studies

The eligible studies were conducted in five of the nine provinces in South Africa. These included the Eastern Cape (45%), Limpopo (30%), North West (15%) Mpumalanga (5%), and KwaZulu-Natal (5%) provinces (Table 1). The majority of these aforementioned provinces are regarded as predominantly rural-based, which may explain the continuous dependence on woody plants for veterinary needs. In a recent review of the ethnoveterinary plants of South Africa, McGaw et al., [15] indicated a similar distribution pattern relating to the use of medicinal plants as remedies for animals/livestock in South Africa. In Pakistan, a rich ethnoveterinary knowledge was recorded in communities residing in remote areas with limited access to conventional veterinary services, which often forced the inhabitants to rely on the natural resources within their immediate environment to meet the health needs of their livestock [14].

The data collection methods included the use of questionnaires, interviews, field observations, and Rapid Rural Appraisal (Table 1). Data on ethnoveterinary medicine was collected from diverse participants such as farmers, cattle headers, indigenous knowledge holders, and traditional healers. In terms of number, the number of participants ranged from 15 [43] to 180 [33] while about 30% of the studies had no indication of the sample size involved in the ethnobotanical survey. Given that the primary focus of these surveys was not on woody plants, varying portions (11–100%) of woody species were identified in the recorded plants (Table 1). In the survey in the Eastern Cape by Dold and Cocks [34], approximately 40% of the 53 recorded plants with ethnoveterinary value were woody plant species. A similar trend was evident in other ethnoveterinary surveys in the Eastern Cape [35,41,51] and Limpopo [48] provinces. In North West province, the portion of woody plants ranged from 26% [47,52] to 32% [43]. In an attempt to understand the basis for the selection and utilisation of plants by local communities, several theories and hypotheses exist [53]. In the current situation, the ecological apparency hypothesis likely accounts for the use of woody plants for ethnoveterinary medicine among local communities. Even though South Africa is recognized as being rich in biodiversity and diverse vegetation-types, it remains highly susceptible due to rapid development, habitat loss, and overexploitation [16]. Increasing evidence supported the dynamic nature of the existing vegetation in South Africa, which is associated with the effects of climate change [54]. On this basis, it is often difficult to predict the pattern for the use of woody plants in ethnoveterinary medicine.

### 3.2. Inventory of Woody Plants with Ethnoveterinary Uses

The high reliance on plants for managing livestock/animals among local communities, especially in developing countries, remains a common trend [14,55]. This popularity has often been attributed to the limited access to convention veterinary drugs and the existence of vast indigenous knowledge for managing livestock in rural communities [47,56,57]. Several studies have revealed that traditional medicines are mostly used because they are regarded as effective and readily available as well as accessible. As often indicated by participants in ethnoveterinary surveys, dependence on traditional medicines is common because western veterinary facilities are inaccessible and are too costly for resource-poor livestock farmers [34,37,45].

Based on the 20 eligible studies from the literature, we generated 104 woody plants with diverse ethnoveterinary uses in South Africa (Table 2). *Terminalia sericea* Burch. Ex DC and *Ziziphus mucronata* Willd were the most common plants, with six mentions. Furthermore, *Cussonia spicata* Thunb., *Pterocarpus angolensis* DC., and *Vachellia karroo* (Hayne) Banfi & Galasso (five citations) and *Cassia abbreviata* Oliv. and *Strychnos henningsii* Gilg (four citations) were popular within the 20 analysed studies from the literature. On the other hand, the majority (86%, i.e., 89 plants) of the 104 woody plants had limited (1–2) mentions.

In terms of plant families (Figure 1), most of the identified plants were from the Fabaceae (19%), Apocynaceae (5.8%), Rubiaceae (5.8%), Anacardiaceae (4.8%), Combretaceae (4.8%), Euphorbiaceae (4.8%), Malvaceae (4.8%), Rhamnaceae (4.8%), and Celastraceae (3.8%) families. Even though 44 families were recorded, the majority (estimated 64%) of the families were represented by one woody plant. Based on the analysis of approximately 4576 vascular plants representing 192 families (from the 254 African families) with medicinal value in sub-Saharan African, the dominance of Fabaceae remains evident in African traditional medicine [58]. Furthermore, Fabaceae was the most represented plant family for plants used against cattle diseases in South Africa [59].

Plant parts used to prepare herbal remedies included bark, leaves, fruits, roots, and flowers (Figure 2). However, the most commonly used plant parts for remedy preparations were bark (33%) followed by leaves (29%) and roots (19%). The dominance of plant parts such as bark and roots may not be sustainable overtime, as their indiscriminate harvesting is often of great conservation concerns for the survival of the plant [26]. Thus, conscious efforts remain essential to ensure good harvesting practices and the long-term sustainability of these valuable woody plants.

### 3.3. Overview of Animals/Livestock and Diseases

As shown in Figure 3, cattle were the major (61%) animal/livestock treated with the woody plants. In South Africa, the importance of cattle among different cultural groups cannot be overemphasized [1,59]. Van der Merwe et al., [52] documented the use of ethnoveterinary medicinal plants in cattle by the Setswana-speaking people in the Madikwe area of the North West Province. The most important diseases treated were retained placenta, diarrhoea, fractures, fertility enhancement, general gastrointestinal problems, and pneumonia. A high proportion of the woody plants were used for diarrhoea. Some of the plants documented during the study are used elsewhere in the Eastern Cape to treat different livestock diseases. These include *Vachellia karroo*, *Vachellia tortilis*, *Cussonia spicata*, *Rhoicissus tridentata*, and *Ziziphus mucronata*.

### 3.4. In Vitro Biological Screening of Woody Plants

The increasing incidence of drug resistance in most pathogenic bacteria and parasites that cause economic loss in animals/livestock production calls for the development of new sources for medication [60,61]. Among to the 104 woody plants with ethnoveterinary uses in South Africa (Table 2), approximately 20% were screened for their relevant biological activities (e.g., antibacterial, anthelmintic, and antioxidant) and safety (cytotoxicity) level. However, the current review included woody plants that have been screened for biological activities without evidence of their ethnoveterinary use in South Africa. This approach may increase the success rate of bio-prospecting for therapeutic woody plants for ethnoveterinary needs in South Africa [62]. As highlighted by Eloff [63], no statistically significant difference was observed in the antimicrobial activity of plants with ethnobotanical knowledge when compared to randomly selected plants. Hence, the most promising biological activity may not correlate with the most popular plants with existing ethnobotanical knowledge [62,64].

#### 3.4.1. Antibacterial Activity

Even though the antibacterial effects of 39 woody plants were reported (Table 3), approximately 56% of the 39 plants lacked ethnoveterinary applications in the eligible studies that were recorded (Table 1 and Table 2). In terms of the assay-type, approximately 95% of the studies were conducting using the micro-plate dilution method, which is considered as a more robust and reliable assay relative to agar diffusion [63,64]. Based on the recorded antibacterial activity (Table 3), Gram-positive bacteria were more dominant (57% of the 14 organisms) than Gram-negative bacterial strains. Although a diverse range of bacterial strains was tested, the relevance and justification for their selection were unclear in most of the studies. Researchers need to be cognizant of the bacteria type in order to demonstrate the clinical relevance of the anti-bacterial effect of the tested plant extracts [64,65].

On the basis on the number of reports, five woody plants namely *Alsophila dregei* (Kunze) R.M.Tryon, *Cussonia spicata* Thunb, *Indigofera frutescens* L.f., *Leucosidea sericea* Eckl. and Zeyh, and *Maesa lanceolata* Forssk were the most studied woody plants in terms of their antibacterial effects (Table 3). The most noteworthy (MIC = 20–40 μg/mL) antibacterial effect (exerted against *Bacillus anthracis*) was demonstrated by acetone extracts of *Bolusanthus speciosus* (Bolus) Harms, *Morus mesozygia* Stapf, and *Maesa lanceolata* Forssk [66]. Likewise, *Salmonella typhimurium* was highly susceptible (MIC = 40 μg/mL) to acetone extracts from *Crotalaria capensis* leaves [67]. Furthermore, the acetone extract from *Maesa lanceolata* leaves exerted a broad-spectrum antibacterial effect by significantly (MIC = 160–630 μg/mL) inhibiting both Gram-positive (*Enterococcus faecalis*, *Staphylococcus aureus*) and Gram-negative (*Escherichia coli*, *Pseudomonas aeruginosa*, *Salmonella typhimurium*) bacterial strains [68]. Similar broad-spectrum antibacterial activity was demonstrated by the acetone leaf extracts of *Indigofera frutescens* L.f. (MIC = 80–310 μg/mL) and *Leucosidea sericea* (MIC = 20–80 μg/mL), as indicated by different authors [67,68,69].

Leaves/aerial parts (77%) and bark (17%) were the most common parts of the woody plants that were evaluated for their antibacterial activity. Remarkable differences in the antibacterial effect of woody plant parts were evident in *Leucosidea sericea* [69], *Schotia brachypetala* Sond, *Searsia lancea* (L.f.) F.A.Barkley (*Rhus lanceas*), and *Ziziphus mucronata* Willd [70]. Particularly in *Schotia brachypetala* and *Ziziphus mucronata*, the leaf extracts had remarkable antibacterial effects against the tested bacterial strains while the bark extracts were ineffective. Furthermore, the type of solvent used for extracting the plant parts strongly influenced the resultant antibacterial response (Table 3). Despite the popularity of water as the most commonly used solvent in traditional medicine, water extracts often exhibit weaker antibacterial effects relative to many organic solvents [63].

#### 3.4.2. Anthelmintic Activity

As highlighted by Aremu et al., [74], evaluating anthelmintic potential is often conducted using (i) developmental and behavioural assays (DBA) and (ii) colorimetric assays (CA). Following treatment and incubation with plant extracts, the assays measure the survival and/or reproductive potential (DBA) or metabolic response using the appropriate marker (CA). A total of 48 woody plants have been tested for their anthelmintic activity, which was mainly (90%) assessed using DBA (Table 4). However, only 42% of these woody plants had existing indigenous knowledge related to the management of animal health among local communities in South Africa. *Alsophila dregei* (Kunze) R.M.Tryon, *Leucosidea sericea*, and *Sclerocarya birrea* were identified as the most commonly evaluated woody plants in terms of their anthelmintic effect. Using pre-defined anthelmintic effect categories [69], the organic solvent extracts of *Leucosidea sericea* had high (minimum lethal concentration, MLC = 0.26–0.52 mg/mL) anthelmintic activity against *Caenorhabditis elegans* [69]. Likewise, Fouche et al., [75] demonstrated that *Maerua angolensis* stem extract exerted 65% inhibition, which was noteworthy among all of the evaluated woody plants. Furthermore, extracts from *Heteromorpha trifoliata*, *Maesa lanceolate*, and *Leucosidea sericea* using an egg hatch assay (e.g., of DBA) exhibited significant anthelmintic activity against *Haemonchus contortus* and killed 100% of the parasites when administered at the dosages of 12.50, 6.25, and 3.13 mg/mL [76]. Fouche et al., [75] investigated the acetone extracts of various woody plants for their anthelmintic activity against *Haemonchus contortus*, and the stem of *Maerua angolensis* had a mean inhibition rate of 65%, which was noteworthy compared to the other plants tested and included in the review.

*Caenorhabditis elegans* (63%), *Haemonchus contortus* (35%), and *Trichostrongylus colubriformis* (2%) have been the widely used organisms for assessing the anthelmintic effects of woody species (Table 4). In recent times, the use of free-living nematodes, particularly, have remained common due to their inherent benefits [74,77,78]. *Caenorhabditis elegans* is regarded as the best representative of a large phylum that contains several parasites [78]. However, the use of *Caenorhabditis elegans* as a test organism has resulted in limited success in terms of the discovery of valuable new leads [62,77,79]. Hence, *Caenorhabditis elegans* should only serve as a screening tool for the rapid identification of promising plant extracts that will be further subjected to more appropriate test model(s).

The type of solvents used for plant extraction has a critical influence on the anthelmintic effect of woody plants (Table 4). For instance, the ethyl acetate extract of *Combretum apiculatum* exhibited strong lethality, killing 70–80% of nematodes (*Caenorhabditis elegans*) while the water extract had a 10–20% killing rate at 1 mg/mL [80]. Furthermore, *Searsia lancea* hexane extract had higher (50%) in vitro anthelmintic activity against *Caenorhabditis elegans* than the methanol and water extracts did [70]. The in vitro anthelmintic efficacy of several woody plants against *Caenorhabditis elegans* revealed that ethanol extracts possessed higher anthelmintic activity than water extracts [71].

Contrary to the majority of studies focusing on an single test organism (Table 4), Shai et al., [81] evaluated the anthelmintic activity of *Curtisia dentata* against parasitic (*Trichostrongylus colubriformis* and *Haemonchus contortus*) and the free-living (*Caenorhabditis elegans*) nematodes. The acetone and dichloromethane extracts were active against all of the nematodes at concentrations as low as 160 μg/mL. This finding clearly highlights the anthelmintic potential of *Curtisia dentata*, which requires further experiments, especially in terms of its in vivo response.

#### 3.4.3. Antioxidant Activity

Antioxidants are free radical scavengers and often possess the ability to reverse or repair the damage caused by free radicals in animal cells [82]. Recently, there has been increasing interest in determining the antioxidant potential of plants used for medicinal purposes [83]. It is generally known that damages caused by reactive oxygen species are often a contributing factor to many diseases [84]. As shown in Table 5, the antioxidant potential of the 24 woody plants have mainly been evaluated via in vitro assays including ABTS—2,2′-azinobis-(3-ethylbenzothiazoline-6-sulfonic acid), DPPH—1,1-diphenyl-2-picryl-hydrazyl, and FRAP -ferric reducing antioxidant power. Relative to the inventory in Table 2, only six woody plants show antioxidant activity.

Given that these bio-analytical assays differ in terms of reaction mechanisms, oxidant, and target species as well as reaction conditions [82], it is often beneficial to evaluate plant extracts in multi-assays. Based on the DPPH assay, the most promising (EC_50_ < 5 μg/mL) antioxidant activity was exerted by woody plants such as *Alsophila dregei*, *Apodytes dimidiata*, *Brachylaena discolor*, *Burkea africana*, *Clausena anisata*, *Combretanum zeyheri*, *Millettia grandis*, *Strychnos mitis*, *Volkameria glabra*, and *Zanthoxylum capense*. Similar noteworthy antixodant effects was observed in the ABTS assay for *Burkea africana* and *Combretum zeyheri* [85]. However, moderate antioxidant activity ranging from 68–579 µg/mL was demonstrated among the eight evaluated woody plants. These aforementioned antioxidant tests were in vitro-based, thereby limiting the clinical relevance of the current findings. Hence, it will be pertinent to establish the in vivo antioxidant activity of woody plants with noteworthy response.

#### 3.4.4. Cytotoxicity

The safety of medicinal plants remains essential toward the drive to incorporate these valuable natural resources as part of healthcare for animals. Evidence of the cytotoxicity levels for 39 woody plants were recorded in the current review (Table 6). Approximately 51% of these woody plants have explicit applications in South African ethnoveterinary medicine (Table 2). Particularly, the safety of different organs from three plants namely *Calpurnia aurea*, *Maesa lanceolata*, and *Sclerocarya birrea* were assessed in more than one study [66,70,75,76].

According to the United States National Cancer Institute (NCI), the criteria for the cytotoxicity of crude extracts, extracts with an LC_50_ value that is less than 20 µg/mL are classified as cytotoxic. On this basis, *Apodytes dimidiate*, *Brachylaena discolour*, *Calpurnia aurea*, *Elaeodendron croceum*, *Maesa lanceolata*, and *Strychnos mitis* exerted varying degrees of cytotoxicity (LC_50_ = 3.32–19.9 μg/mL), and caution needs to be taken during their utilisation for ethnoveterinary medicine [66,67,76]. Furthermore, McGaw et al., [70] assessed the cytotoxicity activity of the hexane, methanol, and water extracts of the selected woody plants against the larvae of *Artemia salina* (brine shrimp). From the results, the water extracts from *Searsia lancea* and *Ziziphus mucronata* leaves displayed strong lethality to the tested organism. On the other hand, moderate cytotoxicity was demonstrated by the acetone and water extracts of *Vachellia nilotica* bark against Vero monkey cell assays, and these extracts exhibited toxic effects on the cells with LC_50_ = 33.2 μg/mL and LC_50_ = 27.8 μg/mL, respectively. This was closely-followed by the acetone extracts from *Tetradenia riparia,* leaf with LC_50_ = 51.3 μg/mL [86].

### 3.5. Phytochemical Analysis of Plants Used for Ethnoveterinary Purposes

Phytochemical screening is important when investigating medicinal plants given that bioactive compounds can be responsible for their resultant biological activities [87,88]. In particular, the phenolic compounds in plants serve as defense mechanisms against pathogens and may be explored for therapeutic purposes [89]. The 20 woody plants recorded exhibit a diverse range of phytochemicals (Table 7), an indication of their potential benefits as ethnoveterinary medicine. For instance, 12 selected woody plant extracts had a rich source of phenols that ranged from 100 to 428 mg GAE/g, and *Lippia javanica* had the highest phenolic content while *Englerophytum magaliesmontanum* had the lowest content [85]. In addition, the flavonoid content varied from 6–159 mg QE/g, as contained in *Ehretia rigida* (lowest) and *Leucaena leucocephala* (highest). Olaokun et al., [90] quantified the total phenolic and flavonoid contents in *Curtisia dentata* and *Pittosporum viridiflorum*. The results indicated that *Curtisia dentata* extract yielded the higher phenolics (125.12 mg/g GAE) and flavonoids (27.69 mg/g GAE) compared to the extract from *Pittosporum viridiflorum*.

In recent times, increasing evidence from several studies on polyphenolic compounds from medicinal plants support their biological and pharmaceutical importance in maintaining animal health and overall productivity [91]. For example, betulinic acid and lupeol were successfully isolated from *Curtisia dentata* [81], which is one of the woody plants recorded in our inventory (Table 2). Subsequently, both compounds demonstrated a moderate degree of an anthelmintic effect (200 and 1 000 μg/mL) against parasitic nematodes. However, the relatively higher concentration required for the compounds to be effective limits their clinical relevance as an anthelmintic for livestock.

## 4. Conclusions

The current review entailed an overview of the role and contributions of woody plants in ethnoveterinary medicine in South Africa. We highlighted the richness of South Africa’s flora as a medicinal resource and the effectiveness of woody plants used in ethnoveterinary medicine. *Terminalia sericea* and *Ziziphus mucronata* were the most commonly utilised woody plants based on existing indigenous knowledge. The extensive utilisation of some plant parts (e.g., bark and roots) remain a major concern due to the potential detrimental effects of the indiscriminate harvesting of such parts may have on the survival and sustainability of the woody plants. The majority (80%) of woody plants with indigenous knowledge related to their applications in the management of animal health remain poorly evaluated in terms of their biological efficacy and phytochemical composition. Nevertheless, some of the woody plants (e.g., *Alsophila dregei*, *Cussonia spicata*, *Indigofera frutescens*, *Leucosidea sericea*, and *Maesa lanceolata*) have demonstrated promising biological activities, mainly in antibacterial and anthelminthic assays. Given the pre-dominantly in vitro based assays currently being utilised, there is an urgent need to evaluate woody plants with promising biological effect in appropriate in vivo models. The test organisms need to have direct relevance to prevailing health challenges facing livestock in rural areas where the use of woody plants have been widely documented. In terms of the phytochemical profiles, South African woody plants have a rich pool of chemicals with potential therapeutic effects.

## Figures and Tables

**Figure 1 vetsci-08-00228-f001:**
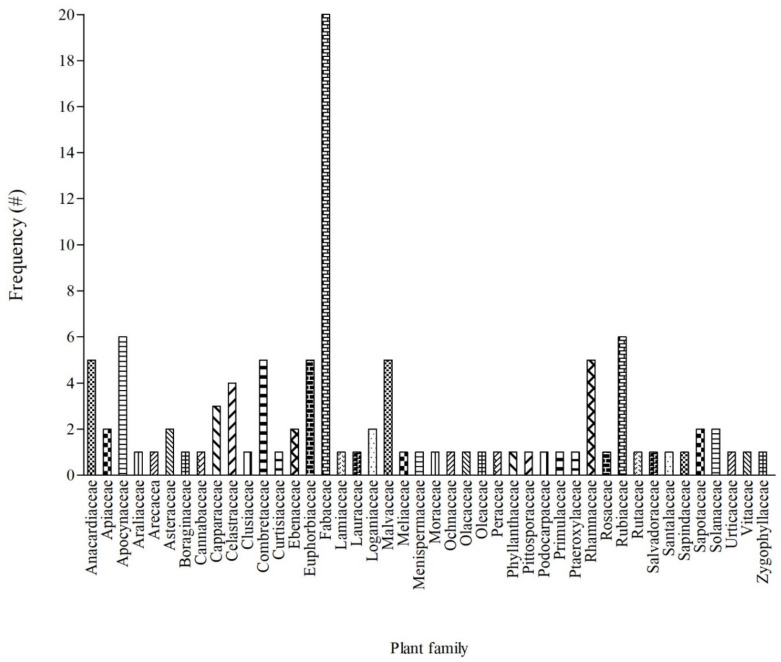
Frequency of the 44 families of woody plants used in South African ethnoveterinary medicine. # = number of mention.

**Figure 2 vetsci-08-00228-f002:**
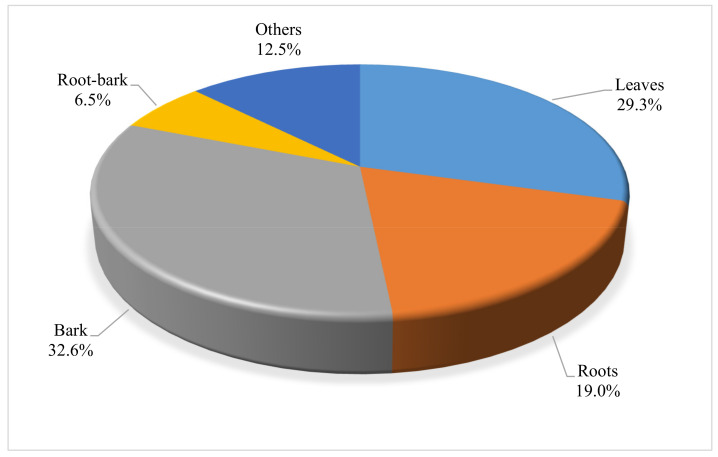
Distribution of different parts of woody plants used in the preparation of ethnoveterinary remedies in South Africa. Others denote parts such as seeds, fruits, flowers, and twigs. (*n* = 184).

**Figure 3 vetsci-08-00228-f003:**
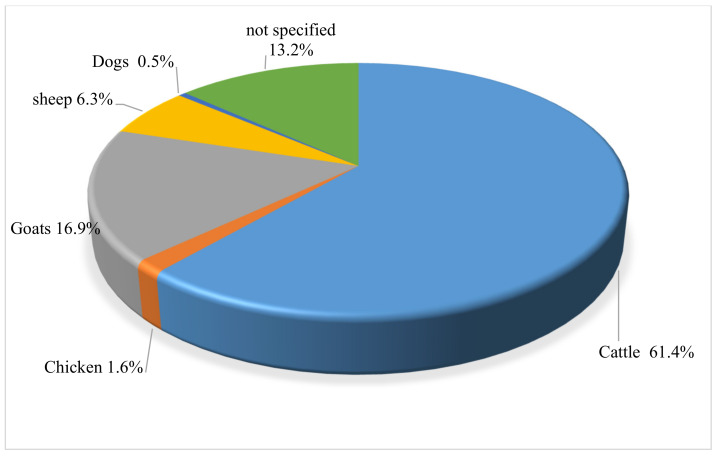
Distribution of animals identified in ethnoveterinary surveys for woody plants in South Africa. (*n* = 189).

**Table 1 vetsci-08-00228-t001:** An overview of ethnoveterinary surveys with evidence on the use of woody plants in South Africa from 2000–2020.

Reference	Province	Documented Plants		Documented Woody Plants		Method of Survey/Interview	Number of Participants
		Number of Plants	Number of Families	Number of Plants	Number of Families		
Chitura et al., [33]	Limpopo	11	10	7	4	Structured questionnaire	180
Dold and Cocks [34]	Eastern Cape	53	42	21	18	Questionnaire and field interview	Not specified
Kambizi [35]	Eastern Cape	22	19	8	7	Semi-structured interview	Not specified
Khunoana et al., [36]	Mpumalanga	11	9	6	4	Semi-structured interview	50
Luseba and Tshisikhawe [37]	Limpopo	34	22	21	13	Focus group discussion	37
Luseba and Van der Merwe [38]	Limpopo	19	12	9	6	Individual and group interview	Not specified
Magwede et al., [39]	Limpopo	27	14	15	9	Open-ended questions and semi-structured questionnaire	42
Mahlo [40]	Limpopo	5	4	5	4	Not specified	Not specified
Maphosa and Masika [41]	Eastern Cape	28	20	12	9	Structured questionnaire and general conversation	30
Mkwanazi et al., [42]	KwaZulu-Natal	5	4	1	1	Structured questionnaire	Not specified
Moichwanetse et al., [43]	North West	25	18	8	5	Face-to-face, semi-structured interview	15
Moyo and Masika [44]	Eastern Cape	2	2	1	1	Structured questionnaire	59
Mthi et al., [45]	Eastern Cape	6	6	4	3	Semi-structured questionnaire, observations and guided field trip	48
Mwale and Masika [46]	Eastern Cape	9	7	1	1	Structured questionnaire	54
Ndou [47]	North West	31	17	8	7	Semi structured face-to-face interview	21
Ramovha and Van Wyk [48]	Limpopo	20	10	14	8	Semi-structured interview	Not specified
Rwodzi [49]	Eastern Cape	10	8	2	2	Questionnaire	60
Sanhokwe et al., [50]	Eastern Cape	9	8	3	3	Structured questionnaire	53
Soyelu and Masika [51]	Eastern Cape	13	11	7	6	Structured questionnaire	53
Van der Merwe et al., [52]	North West	45	24	19	12	Detailed interview	28

**Table 2 vetsci-08-00228-t002:** Inventory of woody plants used for ethnoveterinary purposes among communities in South Africa. Botanical names were validated using PlantZAfrica (http://pza.sanbi.org/ (accessed on 29 September 2021)), The Plant List (http://www.theplantlist.org/ (accessed on 29 September 2021)), and The World Flora Online (http://www.worldfloraonline.org/ (accessed on 29 September 2021)). Syn = synonym; ns = not specified.

Plant Species	Family	Common Name	Method of Preparation and Administration	Plant Part Used	Animal Treated	Disease/Health Condition	Reference
*Acokanthera oppositifolia* (Lam.) Codd	Apocynaceae	Bushman’s Poison	Leaves are boiled for 10 min, strained, and left to stand overnight	Leaves	Sheep, goats	Heartwater	Dold and Cocks [34]
*Acokanthera oppositifolia* (Lam.) Codd	Apocynaceae	Bushman’s Poison	Decoction	Leaves	Goats	Gastrointestinal parasites	Maphosa and Masika [41]
*Acokanthera oppositifolia* (Lam.) Codd	Apocynaceae	Bushman’s Poison	Leaves crushed with water and administered orally	Leaves	Cattle	Paratyphoid (*Goso*)	Mthi et al., [45]
*Acokanthera oppositifolia* (Lam.) Codd	Apocynaceae	Bushman’s Poison	Decoction; ground leaves are boiled, cooled, and administered by drenching the animals. Dose with 1 L bottle for adults and a 300 mL bottle for kids	Leaves	Goats	Helminths, ticks	Sanhokwe et al., [50]
*Afrocarpus falcatus* (Thunb.) C.N.Page	Podocarpaceae	Outeniqua yellowwood	Decoction	Leaves	Dogs	Distemper	Dold and Cocks [34]
*Albizia* sp.	Fabaceae	Xisitana	Root skin is infused in water and is left overnight	Roots	Cattle	Swollen stomach	Khunoana et al., [36]
*Azima tetracantha* Lam.	Salvadoraceae	Beehanger	Dried and ground root is bottled in cold water	Root	Cattle	Dystocia	Dold and Cocks [34]
*Balanites maughamii* Sprague	Zygophyllaceae	Torchwood	Decoction	Leaves	Cattle	Diarrhoea	Mahlo [40]
*Balanites maughamii* Sprague	Zygophyllaceae	Torchwood	Ground leaves are mixed with cold water	Leaves	Cattle	Diarrhoea	Van der Merwe et al., [52]
*Bauhinia thonningii Schum.* (Syn: *Piliostigma* *thonningii* (Schumach.) Milne-Redh.)	Fabaceae	Camel’s foot	Decoction	Leaves	Cattle	Diarrhoea	Mahlo [40]
*Bolusanthus speciosus* (Bolus) Harms	Fabaceae	Tree wisteria	Pounded roots are immersed in water	Roots	Cattle	Retained placenta	Luseba and Tshisikhawe [37]
*Brachylaena ilicifolia* (Lam.) E. Phillips & Schweick.	Asteraceae	Bitterblaar	Leaves are mixed with leaves of *Leucas capensis* (Benth.) Engl. and sap of *Aloe ferox* Mill and boiled	Leaves	Lambs	Diarrhoea	Dold and Cocks [34]
*Breonadia salicina* (Vahl) Hepper & J.R.I.Wood	Rubiaceae	Transvaal teak	Maceration	Bark	Cattle	General intestinal diseases and retained placenta	Mahlo [40]
*Burchellia bubaline* (L.f.) Sims	Rubiaceae	Wild pomegranate	ns	Leaves	ns	Heartwater	Kambizi [35]
*Cadaba aphylla* (Thunb.) Wild	Capparaceae	leafless cadaba, leafless wormbush, black storm	Root decoction: Combined with roots of *Ziziphus zeyheriana*, *Senna italica,* and *Dicoma galpinii*	Root	ns	Blood cleansing and pains (sores, fractures)	Ndou [47]
*Calpurnia aurea* (Aiton) Benth.	Fabaceae	Common calpurnia	Infusion	Leaves	Cattle	Maggot-infested wounds	Soyelu and Masika [51]
*Capparis sepiaria* L.	Capparaceae	Cape Capers	Infusion	Roots	Goats	Gastro-intestinal parasites	Maphosa and Masika [41]
*Carissa bispinosa* (L.) Desf. ex Brenan	Apocynaceae	Forest num-num	Bulb is ground and mixed with water	Roots, bulb	Cattle	Calving difficulties	Luseba and Tshisikhawe [37]
*Cassia abbreviata* Oliv.	Fabaceae	Sjambok pod	Bark infusion	Bark	Cattle	Retained placenta	Chitura et al., [33]
*Cassia abbreviata* Oliv.	Fabaceae	Sjambok pod	Ground bark is soaked in water overnight or boiled	Bark	ns	Worm infestation	Luseba and Van der Merwe [38]
*Cassia abbreviata* Oliv.	Fabaceae	Sjambok pod	Ground bark is mixed with water	Bark	Cattle	Wounds	Magwede et al., [39]
*Cassia abbreviata* Oliv.	Fabaceae	Sjambok pod	Bark infusion or decoction	Bark, root bark	Cattle	Redwater	Ramovha and Van Wyk [48]
*Cassine aethiopica* Thunb. (Syn: *Mystroxylon aethiopicum* (Thunb.) Loes.)	Celastraceae	Kooboo-berry	Bark is grated and boiled for 20 min	Bark	Cattle	Heartwater	Dold and Cocks [34]
*Centella asiatica* (L.) Urb.	Apiaceae	Varkoortjie	Decoction	Bark	Goats	Helminths	Sanhokwe et al., [50]
*Cephalanthus natalensis* Oliv.	Rubiaceae	Strawberry bush	Infusion	Leaves	Cattle	Eye problem	Luseba and Tshisikhawe [37]
*Cissampelos capensis* L.f.	Menispermaceae	Davidjies	ns	Roots	ns	Skin problems, wounds	Kambizi [35]
*Clutia pulchella* L.	Peraceae	Lightning bush	Decoction	Roots	Cattle	Gall	Khunoana et al., [36]
*Coddia rudis* (E.Mey. ex. Harv.) Verdc.	Rubiaceae	Small bone apple	ns	Leaves	Leaves	Skin problems (eliminates ticks)	Kambizi [35]
*Combretum collinum* Fresen	Combretaceae	Bicoloured bushwillow	ns	Bark	Cattle	Constipation	Chitura et al., [33]
*Combretum microphyllum* Klotzsch.	Combretaceae	Flame creeper	Infusion or decoction	Roots	Cattle	Redwater	Ramovha and Van Wyk [48]
*Combretum molle* R.Br ex G.Don	Combretaceae	Velvet bushwillow	Infusion	Leaves	Cattle	Gut conditions—diarrhoea. Worm infestation. Breeding problems, difficult calving	Luseba and Tshisikhawe [37]
*Combretum paniculatum* Vent.	Combretaceae	Burning bush	Decoction	Root bark	Cattle	For fertility problems	Luseba and Van der Merwe [38]
*Croton gratissimus* Burch. (Syn: *Croton gratissimus* Burch. var *gratissimus*)	Euphorbiaceae	lavender croton, lavender fever berry	Dried leaves are crushed and mixed with supplement feed	Leaves	ns	Fertility enhancement in livestock	Ndou [47]
*Croton gratissimus* Burch. (Syn: *Croton gratissimus* Burch. var *gratissimus*)	Euphorbiaceae	Lavender fever berry	ns	Leaves	Cattle	Pneumonia	Van der Merwe et al., [52]
*Curtisia dentata* (Brum. f.) C.A.Smith	Curtisiaceae	Assegai	Bark, together with the bark of *Rapanea melanophloeos* (L.) Mez, is boiled for 30 min	Bark	Cattle	Heartwater	Dold and Cocks [34]
*Cussonia spicata* Thunb.	Araliaceae	Cabbage-tree	Mixed with leaves of *Olea europaea* L. subsp. *africana* (Mill.) P.S.Green to produce concoction or decoction	Leaves	Cattle	Bloody urine after calving (endometritus and/or vaginitis)	Dold and Cocks [34]
*Cussonia spicata* Thunb.	Araliaceae	Cabbage-tree	ns	Bark	ns	Heartwater	Kambizi [35]
*Cussonia spicata* Thunb.	Araliaceae	Cabbage-tree	Infusion	Bark	Goats	Gastro-intestinal parasites	Maphosa and Masika [41]
*Cussonia spicata* Thunb.	Araliaceae	Cabbage-tree	Ground bark is soaked overnight and dose at 300 mL	Bark	Goats	Helmenthis	Sanhokwe et al., [50]
*Cussonia spicata* Thunb.	Araliaceae	Cabbage-tree	ns	Bark	Cattle	Treat retained afterbirth.	Van der Merwe et al., [52]
*Dalbergia obovata* E.Mey.	Fabaceae	Climbing flat bean	Leaves and bark crushed and mixed with water	Leaves, bark	Cattle	Paratyphoid (*Goso*)	Mthi et al., [45]
*Dichrostachys cinerea* (L.) Wight & Arn.	Fabaceae	Sicklebush	Dried fruit is made into powder	Fruit	Sheep, goats	Wounds	Chitura et al., [33]
*Diospyros lycioides* Desf. (Syn: *Diospyros lycioides* Desf. subsp. *lyciodes*)	Ebenaceae	Bluebush, Karoo blue bush	Ground leaves are mixed with water and apply on the affected area	Leaves	Cattle	Ticks	Luseba and Tshisikhawe [37]
*Diospyros lycioides* Desf. (Syn: *Diospyros lycioides* Desf. subsp. *lyciodes*)	Ebenaceae	Bluebush, Karoo blue bush	Leaves are crushed and mixed with water	Leaves	Cattle	Wounds	Magwede et al., [39]
*Diospyros mespiliformis* Hochst. ex A.DC.	Ebeneceae	African ebony	Ground bark is mixed with hippopotamus fat; dosed and also rubbed into vagina	Bark	ns	For milk production	Luseba and Van der Merwe [38]
*Diospyros mespiliformis* Hochst. ex A.DC.	Ebeneceae	African ebony	Ground roots are mixed with warm but not boiling water to yield an infusion	Roots	Cattle	Redwater	Ramovha and Van Wyk [48]
*Dombeya rotundifolia* (Hochst.) Planch.	Malvaceae	Wild pear	Ground leaves/flowers are mixed with chicken feed	Leaves, flowers	Chicken	Newcastle disease	Luseba and Van der Merwe [38]
*Dombeya rotundifolia* (Hochst.) Planch.	Malvaceae	Wild pear	Decoction	Leaves	Cattle	Diarrhoea	Mahlo [40]
*Ehretia rigida* (Thunb.) Druce	Boraginaceae	Puzzle bush	Decoction	Roots	Cattle	Eating problems	Luseba and Tshisikhawe [37]
*Ehretia rigida* (Thunb.) Druce	Boraginaceae	Puzzle bush	ns	Roots	Cattle	Fractures	Van der Merwe et al., [52]
*Elaeodendron transvaalense* (Burtt Davy) R.H.Archer	Celastraceae	Bushveld saffron	Ground fruits are mixed with water	Fruit	Cattle	Worms	Luseba and Tshisikhawe [37]
*Elaeodendron transvaalense* (Burtt Davy) R.H.Archer	Celastraceae	Spike-Thorn	ns	Bark	Cattle	Diarrhoea	Van der Merwe et al., [52]
*Elephantorrhiza burkei* Benth.	Fabaceae	Elephant-root	Ground bulb (or bark) is mixed with water	Bark, roots	Cattle	Diarrhoea	Luseba and Tshisikhawe [37]
*Englerophytum magalismontanum* (Sond.) T.D.Penn	Sapotaceae	Transvaal milkplum	ns	Roots	Cattle	Fertility enhancement	Van der Merwe et al., [52]
*Erythrina caffra* Thunb.	Fabaceae	Coast coral tree	ns	Bark	ns	Heartwater	Kambizi [35]
*Erythrina lysistemon* Hutch.	Fabaceae	Common coral tree	Fresh bark is crushed into pulp and juice is applied	Bark	Cattle	Wounds	Magwede et al., [39]
*Euphorbia cupularis* Boiss.	Euphorbiaceae	Crying tree	Milky latex is applied on third eyelid and on the skin of the limping leg	Milky latex	ns	Eye infection and blackquarter	Luseba and Van der Merwe [38]
*Euphorbia umbellata* (Pax) Bruyns	Euphorbiaceae	African milk bush	Milky sap applied directly on the area between the eye and ear	Stem	Cattle	Eye problem	Khunoana et al., [36]
*Ficus* sp.	Moraceae	ns	ns	Bark	ns	Wounds	Kambizi [35]
*Garcinia livingstonei* T. Anderson	Clusiaceae	African mangosteen	Juice from fresh leaves is squeezed	Leaves	Cattle	Eye problems	Luseba and Tshisikhawe [37]
*Grewia damine* Gaertn. (Syn: *Grewia bicolor* Juss.)	Malvaceae	White raisin	Stem branches are cut into sticks used as lashes	Sticks	Cattle	Redwater	Ramovha and Van Wyk [48]
*Grewia flava* DC.	Malvaceae	Brandybush, wild currant	Root decoction combined with root of *Ziziphus zeyheriana* and given orally	root	cattle	Diarrhoea	Ndou [47]
*Grewia flava* DC.	Malvaceae	Brandybush, wild currant	ns	Roots	Cattle	Fertility enhancement	Van der Merwe et al., [52]
*Grewia occidentalis* L.	Malvaceae	Crossberry	Infusion is prepared with the leaves of *Olea europaea* subsp. *africana* and *Zanthoxylum capense* and sap of *Aloe ferox*	Leaves	ns	Gallsickness	Dold and Cocks [34]
*Grewia occidentalis* L.	Malvaceae	Crossberry	Decoction	Bark	Goats	Gastro-intestinal parasites	Maphosa and Masika [41]
*Grewia occidentalis* L.	Malvaceae	Crossberry	Infusion	Leaves twigs	Cattle	Wounds	Soyelu and Masika [51]
*Gymnosporia* sp.	Celastraceae	Xihlangwa	Root skin infused in water and left overnight	Roots	Cattle	Black quarter and diarrhoea	Khunoana et al., [36]
*Harpephyllum caffrum* Bernh.	Anacardiaceae	Wild plum	ns	Bark	ns	Skin problems	Kambizi [35]
*Harpephyllum caffrum* Bernh.	Anacardiaceae	Wild plum	Decoction	Bark	Goats	Gastro-intestinal parasites	Maphosa and Masika [41]
*Heteromorpha arborescens* (Spreng.) Cham. & Schltdl.	Apiaceae	Parsley tree	Ground root powder is mixed with cold or warm water to yield an infusion	Root	Cattle	Redwater	Ramovha and Van Wyk [48]
*Hippobromus pauciflorus* (L.f) Radlk.	Sapindaceae	Bastard horsewood	Bark is mixed with the bark of *Protorhus longifolia* and is grated and boiled for 10 min	Bark	Cattle	Heartwater and diarrhoea	Dold and Cocks [34]
*Hippobromus pauciflorus* (L.f) Radlk.	Sapindaceae	Bastard horsewood	Infusion	Leaves	Cattle	Wounds	Soyelu and Masika [51]
*Holarrhena pubescens* Wall. ex G.Don	Apocynaceae	Conessi	Crushed roots are mixed with hot water to yield an infusion or are cooked to produce a decoction	Root	Cattle	Redwater	Ramovha and Van Wyk [48]
*Hyperacanthus amoenus* (Sims) Bridson	Rubiaceae	Thorny gardenia	Maceration	Bark	Cattle	Relieving pain, loss of appetite, and general ailments	Mahlo [40]
*Jatropha curcas* L.	Euphorbiaceae	Physic nut	Crushed (1–2) seeds are mixed with water for drenching	Seeds	Cattle, goats	Constipation	Luseba and Van der Merwe [38]
*Jatropha curcas* L.	Euphorbiaceae	Physic nut	Sliced root is cooked to produce a decoction	Root, tuber	Cattle	Redwater	Ramovha and Van Wyk [48]
*Maerua angolensis* DC.	Capparaceae	Bead-bean	Ground leaves are mixed with water. Fresh leaves are squeezed to extract juice	Leaves	Cattle	Eating disorder, drought tonic, eye problems, wounds	Luseba and Tshisikhawe [37]
*Maytenus peduncularis* (Sond.) Loes.	Celastraceae	Blackwood	Root-bark is made into a paste	Root-bark	Cattle	Fractures	Chitura et al., [33]
*Maytenus peduncularis* (Sond.) Loes.	Celastraceae	Blackwood	ns	Leaves	Goats	Ticks	Mkwanazi et al., [42]
*Millettia grandis* (E.Mey) Skeels	Fabaceae	Umzimbeet	Soak the leaves in cold water	Leaves	Chicken	Internal parasites	Mwale and Masika [46]
*Noltea africana* (L.) Rchb. f.	Rhamnaceae	Soap bush	Ground into powder	Roots	Goats	Womb cleansing; fertility	Rwodzi [49]
*Ochna holstii* Engl.	Ochnaceae	Common forest ochna	Leaves and branches boiled for 2 h, 1 litre is given once daily for 3 days	Leaves	Goats, sheep, cattle	Wounds	Luseba and Tshisikhawe [37]
*Ochna holstii* Engl.	Ochnaceae	Common forest ochna	Leaves are ground and boiled	Leaves, twigs, bark	Cattle	Wounds	Magwede et al., [39]
*Ocotea bullata* (Burch.) E. Meyer in Drege	Lauraceae	Stinkwood	Decoction	Bark	Goats	Gastrointestinal parasites	Maphosa and Masika [41]
*Olea europaea* subsp. *cuspidata* (Wall. & G.Don) Cif. (Syn: *Olea europaea* L. subsp. *africana* (Mill.) P.S.Green)	Oleaceae	Wild olive	Bark infusion. Leaves together with *Cussonia spicata* root. Mixture of *Zanthoxylum capense* leaves, *Grewia occidentalis* leaves, and *Aloe ferox* sap	Bark, leaves	Goats, cattle	Diarrhoea in goats. Bloody urine after calving (endomitritis and vaginitis in cattle). Treating gallsickness in cattle	Dold and Cocks [34]
*Olea europaea* subsp. *cuspidata* (Wall. & G.Don) Cif. (Syn: *Olea europaea* L. subsp. *africana* (Mill.) P.S.Green)	Oleaceae	Wild olive	Crushed bark is soaked in warm water	Bark	Cattle	Black quarter (*Ciko*)	Mthi et al., [45]
*Osyris lanceolata* Hoscht. & Steud	Santalaceae	Rock tannin-bush	Maceration	Bulb	Cattle	Retained placenta, alleviation of pain and internal bleeding	Moichwanetse et al., [43]
*Ozoroa paniculosa* (Sond.) R.Fern. & A.Fern. (Syn: *Ozoroa paniculosa* (Sond.) R.Fern. & A.Fern. var. *paniculosa*)	Anacardiaceae	Common resin tree	ns	Bark, root bark	Cattle	Diarrhoea, redwater, sweating sickness	Van der Merwe et al., [52]
*Peltophorum africanum* Sond.	Fabaceae	Weeping wattle	Bark is ground into powder	Bark	Cattle	Wounds	Magwede et al., [39]
*Peltophorum africanum* Sond.	Fabaceae	Weeping wattle	Poultice	Leaves and bulb	Cattle	Retained placenta, diarrhoea, and removal of blood clots from the skin	Moichwanetse et al., [43]
*Peltophorum africanum* Sond.	Fabaceae	Weeping wattle	ns	Bark, root bark	Cattle	Tonic, diarrhoea	Van der Merwe et al., [52]
*Philenoptera violacea* (Klotzsch) Schrire	Fabaceae	Apple-leaf	Bark is ground and infused in water overnight	Bark	Cattle	Gall, diarrhoea, and general ailments	Khunoana et al., [36]
*Philenoptera violacea* (Klotzsch) Schrire	Fabaceae	Apple-leaf	Bark is boiled in water	Bark	Cattle	Wounds	Magwede et al., [39]
*Philenoptera violacea* (Klotzsch) Schrire	Fabaceae	Apple-leaf	Bark is cooked or soaked in cold water to produce a red decoction/infusion	Stem and root-bark	Cattle	Redwater	Ramovha and Van Wyk [48]
*Phoenix reclinata* Jacq.	Arecacea	Wild date palm	Roots are mixed with *Arctotis arctotoides* leaves and boiled, warm liquid is used	Roots	Sheep, goats	Footrot	Dold and Cocks [34]
*Pittosporum viridiflorum* Sims	Pittosporaceae	Cheesewood	Infusion	Bark	Goats	Gastrointestinal parasites	Maphosa and Masika [41]
*Pittosporum viridiflorum* Sims	Pittosporaceae	Cheesewood	Decoction	Roots	Chicken	Wounds	Soyelu and Masika [51]
*Pouzolzia mixta* Solms	Urticaceae	Soap-nettle	Poultice	Roots	Cattle	Retained placenta and uterus cleansing	Moichwanetse et al., [43]
*Protorhus longifolia* (Bernh.) Engl.	Anacardiaceae	Red beech	Mixed with bark of *Hippobromus pauciflorus* and boiled for 20 min	Bark	Cattle	Heartwater and diarrhoea	Dold and Cocks [34]
*Prunus persica* (L.) Batsch	Rosaceae	Peach tree	Decoction	Leaves	Lamb, goats	Diarrhoea	Dold and Cocks [34]
*Prunus persica* (L.) Batsch	Rosaceae	Peach tree	Ground to pulp and mixed with hot paper and liquid	Leaves	ns	Wounds	Magwede et al., [39]
*Prunus persica* (L.) Batsch	Rosaceae	Peach tree	Infusion	Leaves	Cattle	Maggot-infested wounds	Soyelu and Masika [51]
*Pseudolachnostylis maprouneifoloia* Pax	Phyllanthaceae	Kudu berry	Ground bark is mixed with water	Bark	Cattle	Drought tonic	Luseba and Tshisikhawe [37]
*Ptaeroxylon obliquum* (Thunb.) Radlk.	Ptaeroxylaceae	Sneezewood	Decoction	Leaves	Goats	Gastro-intestinal parasites	Maphosa and Masika [41]
*Ptaeroxylon obliquum* (Thunb.) Radlk.	Ptaeroxylaceae	Sneezewood	Crushed and soaked in cold water overnight (infusion)	Bark	Cattle	Ticks	Moyo and Masika [44]
*Ptaeroxylon obliquum* (Thunb.) Radlk.	Ptaeroxylaceae	Sneezewood	Crush bark is mixed with used oil to form paste. Leaf decoction	Bark, leaves	Cattle	Wounds and myiasis	Soyelu and Masika [51]
*Pterocarpus angolensis* DC.	Fabaceae	Paddle-wood	Stem bark infusion	Bark	Cattle	Constipation	Chitura et al., [33]
*Pterocarpus angolensis* DC.	Fabaceae	Paddle-wood	Soak the bark in water	Bark	Cattle	*Mali* and not eating	Luseba and Tshisikhawe [37]
*Pterocarpus angolensis* DC.	Fabaceae	Paddle-wood	Chopped bark is soaked in cold water after the water has changed to reddish boil for 30–60 min	Bark	ns	General illness, unthriftiness, gallsickness, intestinal worms, blackquarter	Luseba and Van der Merwe [38]
*Pterocarpus angolensis* DC.	Fabaceae	Paddle-wood	Bark is ground to pulp	Bark	Cattle	Wounds	Magwede et al., [39]
*Pterocarpus angolensis* DC.	Fabaceae		Bark is cooked or imbibed in cold water to produce a red decoction/infusion	Bark, root bark	Cattle	Redwater	Ramovha and Van Wyk [48]
*Rapanea melanophloeos* (L.) Mez (Syn: *Myrsine melanophloeos* (L.) R.Br. ex Sweet)	Primulaceae	Cape Beech	Mixed with bark of *Curtisia dentata* and boiled for 30 min	Bark	Cattle	Heartwater	Dold and Cocks [34]
*Rauvolfia caffra* Sond.	Apocynaceae	Quinine tree	Applied as powder on wounds	Bark	Cattle	Wounds	Magwede et al., [39]
*Rhamnus prinoides* L’Hér.	Rhamnaceae	Dogwood	Infusion	Roots	Goats	Twin or triplets production	Rwodzi [49]
*Rhoicissus tridentata* (L.f.) Wild & R.B.Drumm.	Vitaceae	Northern bushman’s grape	Tuber is boiled in water for 15 min	Tubers	Goats, sheep	Diarrhoea	Dold and Cocks [34]
*Rhoicissus tridentata* (L.f.) Wild & R.B.Drumm.	Vitaceae	Northern bushman’s grape	Leaves are boiled	Leaves	Cattle	Lumpy skin disease	Luseba and Tshisikhawe [37]
*Rhoicissus tridentata* (L.f.) Wild & R.B.Drumm.	Vitaceae	Northern bushman’s grape	ns	Tubers	Cattle	Heartwater, redwater internal parasites	Van der Merwe et al., [52]
*Rothmannia capensis* Thunb	Rubiaceae	Wild gardenia	Decoction	Roots	Cattle	Eating problem	Luseba and Tshisikhawe [37]
*Rothmannia capensis* Thunb	Rubiaceae	Wild gardenia	Fresh fruits are grounded to pulp	Fruit	ns	Wounds	Magwede et al., [39]
*Schotia brachypetala* Sond.	Fabaceae	African walnut	Ground bark is boiled in water	Bark	Cattle	Foot and mouth diseases, black quarter, and general ailments	Khunoana et al., [36]
*Schotia brachypetala* Sond.	Fabaceae	African walnut	Bark, preferably from the root, is cooked to make a decoction	Bark, root bark	Cattle	Redwater	Ramovha and Van Wyk [48]
*Schotia latifolia* Jacq.	Fabaceae	Bush Boerbean	Decoction	Bark	Goats	Gastro-intestinal parasites	Maphosa and Masika [41]
*Sclerocarya birrea* (A.Rich.) Hochst.	Anacardiaceae	Marula	Bark is soaked in cold water to yield an infusion or is cooked to produce a decoction	Bark	Cattle	Redwater	Ramovha and Van Wyk [48]
*Sclerocarya birrea* (A.Rich.) Hochst.	Anacardiaceae	Marula	ns	Bark	Cattle	Diarrhoea and fractures	Van der Merwe et al., [52]
*Searsia lancea* (L.f.) F.A.Barkley (Syn: *Rhus lancea* L.f.)	Anacardiaceae	Karee	ns	Roots	Cattle	Diarrhoea gallsickness	Van der Merwe et al., [52]
*Secamone filiformis* (L.f) J.H.Ross	Apocynaceae	ns	Stem is ground and mixed with cold water	Stem	Cattle	Diarrhoea	Dold and Cocks [34]
*Senna petersiana* (Bolle) Lock	Fabaceae	Monkey pod	Leaves are soaked	Leaves	Goats	General illnesses	Luseba and Tshisikhawe [37]
*Senna petersiana* (Bolle) Lock	Fabaceae	Monkey pod	Ground root powder is mixed with warm water to yield an infusion	Root	Cattle	Redwater	Ramovha and Van Wyk [48]
*Sideroxylon inerme* L.	Sapotaceae	White milkwood	Bark is crushed and boiled for 20 min	Bark	Cattle	Redwater	Dold and Cocks [34]
*Solanum aculeastrum* Dunal	Solanaceae	Goat bitter-apple	Fresh fruits are ground to pulp	Fruit	ns	Wounds	Magwede et al., [39]
*Spirostachys africana* Sond.	Euphorbiaceae	Tamboti	Bark is ground to pulp	Bark	Cattle	Wounds	Magwede et al., [39]
*Spirostachys africana* Sond.	Euphorbiaceae	Tamboti	ns	Wood	Cattle	Sweating sickness	Van der Merwe et al., [52]
*Strychnos decussata* (Pappe) Gilg.	Loganiaceae	Cape teak	Bark is crushed and soaked in water for 20 min, after which the infusion is strained	Bark	Cattle	Roundworms	Dold and Cocks [34]
*Strychnos henningsii* Gilg	Loganiaceae	Red bitter berry	Resin	ns	Cattle, sheep, goats	Arthritis	Chitura et al., [33]
*Strychnos henningsii* Gilg	Loganiaceae	Red bitter berry	Bark is soaked for 20 min and strained	Bark	Cattle	Heartwater and diarrhoea	Dold and Cocks [34]
*Strychnos henningsii* Gilg	Loganiaceae	Red bitter berry	Decoction	Bark	Goats	Gastro-intestinal parasites	Maphosa and Masika [41]
*Strychnos henningsii* Gilg	Loganiaceae	Red bitter berry	ns	Bark	Cattle	Paratyphoid (*Goso*)	Mthi et al., [45]
*Tabernaemontana elegans* Stapf	Apocynaceae	Toad tree	Crushed or in tact roots are soaked in water to yield an infusion or are cooked to produce a decoction	Roots	Cattle	Redwater	Ramovha and Van Wyk [48]
*Tarchonanthus camphoratus* L.	Asteraceae	Camphor bush	Maceration	Leaves	Cattle	Retained placenta and pain alleviation	Moichwanetse et al., [43]
*Tarchonanthus camphoratus* L.	Asteraceae	Camphor bush	Leaf infusion, oral route: The leaves of the plant are put in drinking water	Leaves	ns	To prevent cold	Ndou [47]
*Terminalia sericea* Burch. ex DC.	Combretaceae	Silver cluster-leaf	Ground roots are mixed with water to apply on the ticks and wounds. Roots are boiled and given to the animal	Roots	Cattle	Ticks and wounds, diarrhoea	Luseba and Tshisikhawe [37]
*Terminalia sericea* Burch. ex DC.	Combretaceae	Silver cluster-leaf	Ground leaves are mixed with water and applied on the wound and are covered with cattle dung	Leaves	Cattle	Wounds	Luseba and Van der Merwe [38]
*Terminalia sericea* Burch. ex DC.	Combretaceae	Silver cluster-leaf	Roots are ground to pulp and mixed with water	Roots	Cattle	Wounds and ticks	Magwede et al., [39]
*Terminalia sericea* Burch. ex DC.	Combretaceae	Silver cluster-leaf	Poultice	Leaves	Cattle	Retained placenta and uterus cleansing	Moichwanetse et al., [43]
*Terminalia sericea* Burch. ex DC.	Combretaceae	Silver cluster-leaf	Root-bark is soaked in cold water to yield an infusion, or dried bark is ground to a powder and is mixed with water	Roots, bark	Cattle	Redwater	Ramovha and Van Wyk [48]
*Terminalia sericea* Burch. ex DC.	Combretaceae	Silver cluster-leaf	ns	Roots	Cattle	Diarrhoea	Van der Merwe et al., [52]
*Trema orientalis* (L.) Blume	Cannabaceae	Pigeonwood	Ground leaves are mixed with water	Leaves	Cattle, goats, sheep	Eye problems and gallsickness	Luseba and Tshisikhawe [37]
*Triumfetta sonderi* Ficalho & Hiern	Malvaceae	Sonder’s truimfetta	ns	Root-bark	Cattle	Retained placenta	Van der Merwe et al., [52]
*Turraea obtusifolia* Hochst.	Meliaceae	Small honeysuckle tree	Crushed leaves are applied directly on the wounds	Leaves	Goats, sheep, cattle	Wounds	Luseba and Tshisikhawe [37]
*Vachellia karroo* (Hayne) Banfi & Galasso (Syn: *Acacia karroo*)	Fabaceae	Sweet thorn	Bark is chopped into small pieces and boiled	Bark	Goats, sheep	Diarrhoea and intestinal parasites	Dold and Cocks [34]
*Vachellia karroo* (Hayne) Banfi & Galasso (Syn: *Acacia karroo*)	Fabaceae	Sweet thorn	Maceration	Bulb	Cattle	Retained placenta and bacterial infections	Moichwanetse et al., [43]
*Vachellia karroo* (Hayne) Banfi & Galasso (Syn: *Acacia karroo*)	Fabaceae	Sweet thorn	Leaves are crushed and mixed with Madubula	Leaves	Cattle	Wounds and myiasis	Soyelu and Masika [51]
*Vachellia karroo* (Hayne) Banfi & Galasso (Syn: *Acacia karroo*)	Fabaceae	Sweet thorn	ns	Bark	Cattle	Fractures and diarrhoea	Van der Merwe et al., [52]
*Vachellia karroo* (Hayne) Banfi & Gallaso (Syn: *Acacia karroo*)	Fabaceae	Sweet thorn	For external coaptation of simple bone fractures (*thobega*)	Thorn, bark	ns	Fracture repair and splints for fracture repair	Ndou [47]
*Vachellia tortilis* (Forssk.) Galasso & Banfi (Syn: *Acacia tortilis*)	Fabaceae	Umbrella thorn	ns	Branch tips	Cattle	Diarrhoea	Van der Merwe et al., [52]
*Volkameria glabra* (E.Mey.) Mabb. & Y.W.Yuan (Syn: *Clerodendrum capense* D.Don ex Steud.)	Lamiaceae	Tinderwood	ns	Leaves	ns	Worms	Kambizi [35]
*Withania somnifera* (L.) Dunal	Solanaceae	winter cherry	Tuber infusion combined with roots of *Solanum lichtensteinii* and *Bulbine abyssinica*, oral route	Tubers	ns	Internal sores	Ndou [47]
*Xanthocercis zambesiaca* (Baker) Dumaz-le-Grand	Fabaceae	Nyala tree	Ground bark is given to cattle for eating disorders. Ground bark is mixed with salt or leaves are soaked for 12 h	Bark, leaves	Cattle	Eating problem and diarrhoea	Luseba and Tshisikhawe [37]
*Xanthocercis zambesiaca* (Baker) Dumaz-le-Grand	Fabaceae	Nyala tree	Ground bark is applied topically	Bark	ns	Wounds	Magwede et al., [39]
*Ximenia americana* L. var. *microphylla* Welw. ex Oliv.	Olacaceae	Tallowwood	Root-bark is powdered	Root bark	Cattle sheep, goats	Wounds	Luseba and Tshisikhawe [37]
*Ximenia americana* L. var. *microphylla* Welw. ex Oliv.	Olacaceae	Tallowwood	ns	Roots	Cattle	Internal parasites	Van der Merwe et al., [52]
*Zanthoxylum capense* (Thunb.) Harv.	Rutaceae	Small knobwood	Infusion prepared *Grewia occidentalis*, *Olea europaea* subsp. *africana* leaves and *Aloe ferox* sap	Leaves	ns	Gallsickess	Dold and Cocks [34]
*Zanthoxylum capense* (Thunb.) Harv.	Rutaceae	Small knobwood	Decoction	Roots	Goats	Gastro-intestinal parasites	Maphosa and Masika [41]
*Ziziphus mucronata* Willd.	Rhamnaceae	Buffalo thorn	Leaf paste	Leaves	Cattle	Mastitis	Chitura et al., [33]
*Ziziphus mucronata* Willd.	Rhamnaceae	Buffalo thorn	Bark is soaked in water while leaves are ground into pulp	Bark and leaves	ns	Wound	Magwede et al., [39]
*Ziziphus mucronata* Willd.	Rhamnaceae	Buffalo thorn	Infusion	Roots	Goats	Gastro-intestinal parasites	Maphosa and Masika [41]
*Ziziphus mucronata* Willd.	Rhamnaceae	Buffalo thorn	Poultice	Roots	Cattle	Retained placenta	Moichwanetse et al., (2020)
*Ziziphus mucronata* Willd.	Rhamnaceae	Buffalo thorn	Crushed leaves and soft branches poultice: crushed and placed on a hard abscess	Leaves, branches	ns	Abscess ripening	Ndou [47]
*Ziziphus mucronata* Willd.	Rhamnaceae	Buffalo thorn	ns	Leaves, roots	Cattle	Fertility enhancement, sores and burns	Van der Merwe et al., [52]
*Ziziphus oxyphylla* Edgew (Syn: *Ziziphus acuminata* Royle)	Rhamnaceae	Pointed-leaf jujube	Poultice	Roots	Cattle	Retained placenta and increase stimulation for separating retained placenta	Moichwanetse et al., [43]
*Ziziphus zeyheriana* Sond.	Rhamnaceae	Dwarf buffalo-thorn	ns	Root-stock	ns	Diarrhoea internal parasites. General ailments	Luseba and Van der Merwe [38]
*Ziziphus zeyheriana* Sond.	Rhamnaceae	Dwarf buffalo-thorn	Root decoction: Combined with roots of *Cadaba aphylla*, *Senna italica* and *Dicoma galpinii*. Root decoction combined with root of *Helichrysum caespititium*. Root decoction combined with root of *Grewia flava*, oral route. The sick calf is given about half a litre of the decoction orally	Roots	Cattle	Blood cleansing, pains (from sores, fractures), calf diarrhoea	Ndou [47]

**Table 3 vetsci-08-00228-t003:** Examples of in vitro antibacterial activity of woody plants with ethnoveterinary applications in South Africa. # Plant species: denotes woody plants with ethnoveterinary uses in Table 2; MIC—minimum inhibitory concentration, ns—not specified.

# Plant Species	Plant Part	Solvent	Test System	Test Organism	Positive Control	Findings	Reference
*Acokanthera oppositifolia*	Leaves	Petroleum ether, dichloromethane, ethanol, and water	Serial microplate dilution	*Bacillus subtilis, Staphylococcus aureus, Escherichia coli* and *Klebsiella pneumoniae*	Neomycin (0.39–1.56 µg/mL)	All extract had no noteworthy (MIC > 1 mg/mL) antibacterial effect	Aremu et al., [71]
*Alsophila dregei* (Kunze) R.M.Tryon (Syn: *Cyathea dregei*)	Leaves	Acetone	Serial microplate dilution	*Escherichia coli*, *Enterococcus* *faecalis*, *Pseudomonas* *aeruginosa,* and *Staphylococcus aureus*	Gentamicin ≤ 0.02 mg/mL	Moderate antibacterial activity with MIC = 0.63 mg/mL	Adamu et al., [68]
*Alsophila dregei* (Kunze) R.M.Tryon (Syn: *Cyathea dregei*)	Leaves, roots	Petroleum ether, dichloromethane, ethanol, and water	Serial microplate dilution	*Bacillus subtilis*, *Staphylococcus aureus*, *Escherichia coli* and *Klebsiella pneumoniae*	Neomycin (0.39–1.56 µg/mL)	Petroleum ether and ethanol root extracts had noteworthy antibacterial activity (MIC < 1 mg/mL) against Gram-positive bacteria	Aremu et al., [71]
*Apodytes dimidiata* E.Mey. ex Arn.	Leaves	Acetone	Serial microplate dilution	*Escherichia coli*, *Enterococcus* *faecalis*, *Pseudomonas* *aeruginosa,* and *Staphylococcus aureus*	Gentamicin ≤ 0.02 mg/mL	Moderate antibacterial activity with MIC = 0.31 mg/mL against *Staphylococcus aureus* and *Pseudomonas aeruginosa*	Adamu et al., [68]
*Baphia racemosa* (Hochst.) Baker	Leaves	Acetone	Serial microplate dilution	*Staphylococcus aureus, Enterococcus faecalis*, *Bacillus cereus, Escherichia coli, Pseudomonas* *aeruginosa,* and *Salmonella typhimurium*	Gentamicin = 0.2–1.56 μg/mL	Noteworthy effect against *Enterococcus faecalis* (MIC = 160 μg/mL) and *Staphylococcus aureus* (MIC = 310 μg/mL)	Dzoyem et al., [67]
*Berchemia zeyheri* (Sond.) Grubov	Bark	Hexane, methanol, and water	Serial microplate dilution method	*Escherichia coli*, *Enterococcus* *faecalis*, *Pseudomonas* *aeruginosa,* and *Staphylococcus aureus*	Neomycin (0.78–25 μM)	*Staphylococcus aureus* was susceptible (MIC < 1 mg/mL) to hexane and methanol extracts	McGaw et al., [70]
*# Bolusanthus speciosus* (Bolus) Harms	Leaves	Acetone	Serial microplate dilution	*Bacillus anthracis*	Gentamicin = 0.0002 mg/mL	MIC = 0.04 mg/mL	Elisha et al., [66]
*# Calpurnia aurea* (Aiton) Benth.	Leaves	Acetone	Serial microplate dilution	*Bacillus anthracis*	Gentamicin = 0.0002 mg/mL	MIC = 0.31 mg/mL	Elisha et al., [66]
*Clausena anisata* (Willd.) Hook.f. ex. Benth.	Leaves	Acetone	Serial microplate dilution	*Escherichia coli*, *Enterococcus* *faecalis*, *Pseudomonas* *aeruginosa,* and *Staphylococcus aureus*	Gentamicin ≤ 0.02 mg/mL	Noteworthy antibacterial activity (MIC = 0.16–0.31 mg/mL)	Adamu et al., [68]
*Combretum caffrum* Eckl. & Zeyh.) Kuntze	Bark	Acetone, methanol, and water	Agar plate	*Escherichia coli*, *Pseudomonas* *aeruginosa*, *Staphylococcus* *aureus*, *Bacillus cereus*, *Bacillus pumilus*, *Bacillus subtilis*, *Micrococcus kristinae*, *Klebsiella* *pneumonia*, *Serratia marcescens,* and *Enterobacter cloacae*	ns	Methanol extract inhibited both Gram-positive and Gram-negative bacteria ranging from 0.5–5 mg/mL. Acetone extract mainly inhibited (MIC = 0.5 mg/mL) Gram-positive bacterial strains. Water extract showed activity against five Gram-positive and one Gram-negative bacteria	Masika and Afolayan [72]
*Cremaspora triflora* (Thonn.) K.Schum.	Leaves	Acetone	Serial microplate dilution	*Bacillus anthracis*	Gentamicin = 0.0002 mg/mL	MIC = 0.16 mg/mL	Elisha et al., [66]
*Crotalaria capensis* Jacq.	Leaves	Acetone	Serial microplate dilution	*Staphylococcus aureus, Enterococcus faecalis*, *Bacillus cereus, Escherichia coli, Pseudomonas* *aeruginosa,* and *Salmonella typhimurium*	Gentamicin = 0.2–1.56 μg/mL	Noteworthy effect against *Enterococcus faecalis* (MIC = 80 μg/mL) and *Salmonella typhimurium* (MIC = 20 μg/mL)	Dzoyem et al., [67]
*# Cussonia spicata* Thunb.	Bark	Methanol and dichloromethane	Serial microplate dilution	*Escherichia coli*, *Pseudomonas* *aeruginosa,* and *Staphylococcus aureus*	Neomycin = 0.l mg/mL	No noteworthy antibacterial activity	Luseba et al., [73]
*# Cussonia spicata* Thunb.	Root	Hexane, methanol, and water	Serial microplate dilution	*Escherichia coli*, *Enterococcus* *faecalis*, *Pseudomonas* *aeruginosa,* and *Staphylococcus aureus*	Neomycin (0.78–25 μM)	No noteworthy (MIC > 1 mg/mL) antibacterial effects for all of the tested extracts	McGaw et al., [70]
*Dalbergia nitidula* Baker	Leaves	Acetone	Serial microplate dilution	*Staphylococcus aureus, Enterococcus faecalis*, *Bacillus cereus*, *Escherichia coli*, *Pseudomonas* *aeruginosa,* and *Salmonella typhimurium*	Gentamicin = 0.2–1.56 μg/mL	Noteworthy effect against *Bacillus cereus* (MIC = 80 μg/mL)	Dzoyem et al., [67]
*# Dombeya rotundifolia* (Hochst.) Planch.	Aerial part	Hexane, methanol, and water	Serial microplate dilution	*Escherichia coli*, *Enterococcus* *faecalis*, *Pseudomonas* *aeruginosa,* and *Staphylococcus aureus*	Neomycin (0.78–25 μM)	Methanol extract had noteworthy (MIC = 0.4 mg/mL) antibacterial effect against Gram-positive bacteria	McGaw et al., [70]
*Elaeodendron croceum* (Thunb.) DC.	Leaves	Acetone	Serial microplate dilution	*Bacillus anthracis*	Gentamicin = 0.0002 mg/mL	MIC = 0.31 mg/mL	Elisha et al., [66]
*Erythrina caffra* Thunb.	Leaves	Acetone	Serial microplate dilution	*Staphylococcus aureus, Enterococcus faecalis*, *Bacillus cereus*, *Escherichia coli*, *Pseudomonas* *aeruginosa,* and *Salmonella typhimurium*	Gentamicin = 0.2–1.56 μg/mL	Noteworthy effect against *Enterococcus faecalis* (MIC = 80 μg/mL)	Dzoyem et al., [67]
# *Euphorbia cupularis* Boiss. *Synadenium cuplare*)	Stem/leaves	Hexane	Serial microplate dilution	*Escherichia coli*, *Enterococcus* *faecalis*, *Pseudomonas* *aeruginosa,* and *Staphylococcus aureus*	Neomycin (0.78–25 μM)	Hexane extract showed a weak inhibition against two Gram- positive	McGaw et al., [70]
*# Heteromorpha* *arborescens* (Spreng.) Cham. & Schltdl.	Leaves	Acetone	Serial microplate dilution	*Bacillus anthracis*	Gentamicin = 0.0002 mg/mL	MIC = 0.16 mg/mL	Elisha et al., [66]
*Heteromorpha trifoliata* (H.L.Wendl.) Eckl. & Zeyh	Leaves	Acetone	Serial microplate dilution	*Escherichia coli*, *Enterococcus* *faecalis*, *Pseudomonas* *aeruginosa,* and *Staphylococcus aureus*	Gentamicin ≤ 0.02 mg/mL	Moderate antibacterial activity with MIC = 0.63 against two Gram-negative bacteria	Adamu et al., [68]
*# Hippobromus* *pauciflorus* (L.f.) Radlk.	Aerial part	Hexane, methanol, and water	Serial microplate dilution	*Escherichia coli*, *Enterococcus* *faecalis*, *Pseudomonas* *aeruginosa,* and *Staphylococcus aureus*	Neomycin (0.78–25 μM)	Methanol extracts had noteworthy antibacterial effect (MIC = 0.2 mg/mL) against *Staphylococcus aureus*	McGaw et al., [70]
*Indigofera frutescens* L.f. (Syn: *Indigofera* *cylindrica* DC.)	Leaves	Acetone	Serial microplate dilution	*Escherichia coli*, *Enterococcus* *faecalis*, *Pseudomonas* *aeruginosa,* and *Staphylococcus aureus*	Gentamicin ≤ 0.02 mg/mL	Noteworthy antibacterial effect (MIC = 0.08–0.31 mg/mL) against the four bacterial strains	Adamu et al., [68]
*Indigofera frutescens* L.f. (Syn: *Indigofera* *cylindrica* DC.)	Leaves	Acetone	Serial microplate dilution	*Staphylococcus aureus, Enterococcus faecalis*, *Bacillus cereus, Escherichia coli, Pseudomonas* *aeruginosa,* and *Salmonella typhimurium*	Gentamicin = 0.2–1.56 μg/mL	Noteworthy effect against *Salmonella typhimurium* (MIC = 40 μg/mL)	Dzoyem et al., [67]
*Leucosidea sericea* Eckl. & Zeyh.	Leaves	Acetone	Serial microplate dilution	*Escherichia coli*, *Enterococcus* *faecalis*, *Pseudomonas* *aeruginosa,* and *Staphylococcus aureus*	Gentamicin ≤ 0.02 mg/mL	Noteworthy antibacterial effect (MIC = 0.02–0.08 mg/mL) against the four bacterial strains	Adamu et al., [68]
*Leucosidea sericea* Eckl. & Zeyh.	Leaves, stem	Petroleum ether, dichloromethane, ethanol, water	Serial microplate dilution	*Bacillus subtilis*, *Staphylococcus aureus*, *Escherichia coli*, and *Klebsiella pneumonia*	Neomycin (0.39–1.56 μg/mL)	Majority of the solvent extracts from the leaves had noteworthy antibacterial effect (MIC = 0.025–0.78 mg/mL) against all four bacterial strains. Stem organic solvent extracts had remarkable MIC (0.39–0.78 mg/mL) against Gram-positive bacteria	[69]
*Lonchocarpus nelsii* (Schinz) Heering & Grimme	Leaves	Acetone	Serial microplate dilution	*Staphylococcus aureus, Enterococcus faecalis*, *Bacillus cereus, Escherichia coli, Pseudomonas* *aeruginosa,* and *Salmonella typhimurium*	Gentamicin = 0.2–1.56 μg/mL	Noteworthy effect against *Enterococcus faecalis* and *Salmonella typhimurium* (MIC = 80 μg/mL)	Dzoyem et al., [67]
*Maesa lanceolata* Forssk.	Leaves	Acetone	Serial microplate dilution	*Escherichia coli*, *Enterococcus* *faecalis*, *Pseudomonas* *aeruginosa,* and *Staphylococcus aureus*	Gentamicin ≤ 0.02 mg/mL	Noteworthy antibacterial effect (MIC = 0.02–0.08 mg/mL) against the four bacterial strains	Adamu et al., [68]
*Maesa lanceolata* Forssk.	Leaves	Acetone	Serial microplate dilution	*Bacillus anthracis*	Gentamicin = 0.0002 mg/mL	MIC = 0.02 mg/mL	Elisha et al., [66]
*Melia azedarach* L.	Leaves	Acetone	Serial microplate dilution	*Escherichia coli*, *Enterococcus* *faecalis*, *Pseudomonas* *aeruginosa,* and *Staphylococcus aureus*	Gentamicin ≤ 0.02 mg/mL	Noteworthy antibacterial effect (MIC = 0.16–0.63 mg/mL) against the four bacterial strains	Adamu et al., [68]
*# Millettia grandis* (E.Mey.) Skeels	Leaves	Acetone	Serial microplate dilution	*Escherichia coli*, *Enterococcus* *faecalis*, *Pseudomonas* *aeruginosa,* and *Staphylococcus aureus*	Gentamicin ≤ 0.02 mg/mL	Moderate antibacterial activity with MIC = 0.31 mg/mL against four bacteria	Adamu et al., [68]
*Morus mesozygia* Stapf	Leaves	Acetone	Serial microplate dilution	*Bacillus anthracis*	Gentamicin = 0.0002 mg/mL	MIC = 0.04 mg/mL	Elisha et al., [66]
*# Pittosporum viridiflorum* Sims	Leaves	Acetone	Serial microplate dilution	*Bacillus anthracis*	Gentamicin = 0.0002 mg/mL	MIC = 0.08 mg/mL	Elisha et al., [66]
*Podalyria calyptrata* (Retz.) Willd.	Leaves	Acetone	Serial microplate dilution	*Staphylococcus aureus, Enterococcus faecalis*, *Bacillus cereus, Escherichia coli, Pseudomonas* *aeruginosa,* and *Salmonella typhimurium*	Gentamicin = 0.2–1.56 μg/mL	Noteworthy effect against *Salmonella typhimurium* (MIC = 160 μg/mL)	Dzoyem et al., [67]
*# Pterocarpus angolensis* DC.	Bark	Dichloromethane and 90% methanol	Serial microplate dilution	*Escherichia coli*, *Pseudomonas* *aeruginosa,* and *Staphylococcus aureus*	Neomycin = 0.l mg/mL	Moderate antibacterial activity with MIC = 0.31 mg/mL against *Staphylococcus aureus*	Luseba et al., [73]
*Salix mucronata* subsp. *capensis* (Thunb.) Immelman (*Salix capensis*)	Bark	Acetone, methanol, and water	Agar plate	*Escherichia coli*, *Pseudomonas* *aeruginosa*, *Staphylococcus* *aureus*, *Bacillus cereus*, *Bacillus pumilus*, *Bacillus subtilis*, *Micrococcus kristinae*, *Klebsiella* *pneumonia*, *Serratia marcescens,* and *Enterobacter cloacae*	ns	Acetone and methanol extracts inhibited both Gram-positive and Gram-negative bacteria ranging from 0.5 to 5 mg/mL	Masika and Afolayan [72]
*# Schotia brachypetala* Sond.	Bark, leavse	Hexane, methanol, and water	Serial microplate dilution	*Escherichia coli*, *Enterococcus* *faecalis*, *Pseudomonas* *aeruginosa,* and *Staphylococcus aureus*	Neomycin (0.78–25 μM)	Bark methanol extract had noteworthy antibacterial effect (MIC = 0.1–0.2 mg/mL) against two Gram-positive bacterial strains. Leaf methanol extract had noteworthy antibacterial effect (MIC = 0.2–0.4 mg/mL) against two Gram-positive bacterial strains	McGaw et al., [70]
*# Sclerocarya birrea* (A. Rich.) Hochst.	Leaves	Hexane, methanol, and water	Serial microplate dilution	*Escherichia coli*, *Enterococcus* *faecalis*, *Pseudomonas* *aeruginosa,* and *Staphylococcus aureus*	Neomycin (0.78–25 μM)	Methanol extract had noteworthy antibacterial effect (MIC = 0.1–0.4 mg/mL) against two Gram- positive bacterial strains	McGaw et al., [70]
*# Searsia lancea* (L.f.) F.A.Barkley (*Rhus lanceas*)	Bark, leaves	Hexane, methanol, and water	Serial microplate dilution	*Escherichia coli*, *Enterococcus* *faecalis*, *Pseudomonas* *aeruginosa,* and *Staphylococcus aureus*	Neomycin (0.78–25 μM)	Bark methanol extract had noteworthy antibacterial effect (MIC = 0.2 mg/mL) against two Gram-positive bacterial strains. Leaf methanol extract had noteworthy MIC (0.2 mg/mL) against *Staphylococcus aureus*	McGaw et al., [70]
*Virgilia divaricata* Adamson	Leaves	Acetone	Serial microplate dilution	*Staphylococcus aureus, Enterococcus faecalis*, *Bacillus cereus*, *Escherichia coli*, *Pseudomonas* *aeruginosa*, and *Salmonella typhimurium*	Gentamicin = 0.2–1.56 μg/mL	Noteworthy effect against *Bacillus cereus* and *Salmonella typhimurium* (MIC = 80 μg/mL)	Dzoyem et al., [67]
# *Volkameria glabra* (E. Mey.) Mabb. & Y. W. Yuan (Syn: *Clerodendrum glabrum*)	Leaves	Acetone	Serial microplate dilution	*Escherichia coli*, *Enterococcus* *faecalis*, *Pseudomonas* *aeruginosa,* and *Staphylococcus aureus*	Gentamicin ≤ 0.02 mg/mL	Noteworthy antibacterial effect (MIC = 0.31–0.63 mg/mL) against two Gram-negative bacterial strains	Adamu et al., [68]
*Xylia torreana* Brenan	Leaves	Acetone	Serial microplate dilution	*Staphylococcus aureus, Enterococcus faecalis*, *Bacillus cereus*, *Escherichia coli*, *Pseudomonas* *aeruginosa,* and *Salmonella typhimurium*	Gentamicin = 0.2–1.56 μg/mL	Noteworthy effect against *Bacillus cereus* and *Salmonella typhimurium* (MIC = 160 μg/mL)	Dzoyem et al., [67]
# *Zanthoxylum capense* (Thunb.) Harv.	Leaves	Acetone	Serial microplate dilution	*Escherichia coli*, *Enterococcus* *faecalis*, *Pseudomonas* *aeruginosa,* and *Staphylococcus aureus*	Gentamicin ≤ 0.02 mg/mL	Noteworthy antibacterial effect (MIC = 0.31 mg/mL) against *Enterococcus faecalis* and *Pseudomonas aeruginosa*	Adamu et al., [68]
*# Ziziphus mucronata* Willd.	Bark, leaves	Hexane, methanol, and water	Serial microplate dilution	*Escherichia coli*, *Enterococcus* *faecalis*, *Pseudomonas* *aeruginosa,* and *Staphylococcus aureus*	Neomycin (0.78–25 μM)	Bark extracts had no noteworthy antibacterial effect (MIC > 1 mg/mL) against all of the tested bacterial strains. Leaf methanol extracts had noteworthy antibacterial effect (MIC = 0.2 mg/mL) against *Staphylococcus* *aureus*	McGaw et al., [70]

**Table 4 vetsci-08-00228-t004:** Examples of anthelmintic effects of woody plants with ethnoveterinary applications in South Africa. # Plant species: denotes woody plants with ethnoveterinary uses in Table 2; ^$^ Assay type: CA—colourimetric assay, DBA—developmental and behavioral assay; * Findings: EHA—egg hatch assay; LDT—larval development test; MLC—minimum lethal concentration.

# Plant Species	Solvent	^$^ Assay Type	Plant-Part	Parasite	Positive Control	* Findings	Reference
*# Acokanthera oppositifolia* (Lam.) Codd	Petroleum ether, dichloromethane, ethanol, and water	CA	Leaves, twigs	*Caenorhabditis* *elegans*	Levamisole (40 μg/mL)	Petroleum ether and ethanol leaf extracts had noteworthy MLC (0.52 mg/mL)	Aremu et al., [71]
*Alsophila dregei* (Kunze) R.M.Tryon (*Cyathea dregei*)	Petroleum ether, dichloromethane, ethanol, and water	CA	Leaves, roots	*Caenorhabditis* *elegans*	Levamisole (40 μg/mL)	Dichloromethane and ethanol leaf extracts had noteworthy MLC (0.52 mg/mL)	Aremu et al., [71]
*Alsophila dregei* (Kunze) R.M.Tryon (*Cyathea dregei*)	Acetone	DBA	Leaves	*Haemonchus* *contortus*	Albendazole (0.008–25 μg/mL)	EC_50_ *=* 17.64 mg/mL (EHA), 17.93 mg/mL (LDT)	Adamu et al., [76]
*Alsophila dregei* (Kunze) R.M.Tryon (*Cyathea dregei*)	Petroleum ether, dichloromethane, ethanol, and water	CA	Leaves, roots	*Caenorhabditis* *elegans*	Levamisole (40 μg/mL)	Dichloromethane and ethanol extracts had noteworthy MLC (0.52 mg/mL)	Aremu et al., [71]
*Apodytes dimidiata*	Acetone	DBA	Leaves	*Haemonchus* *contortus*	Albendazole (0.008–25 μg/mL)	EC_50_ = 5.7 mg/mL (EHA), 4.13 mg/mL (LDT)	Adamu et al., [76]
*Berchemia zeyheri*	Hexane, methanol, and water	DBA	Bark	*Caenorhabditis* *elegans*	Levamisole (10 μg/mL)	Methanol extract had moderate (30%) lethality at 2 mg/mL	McGaw et al., [70]
*Brachylaena discolor*	Acetone	DBA	Leaves	*Haemonchus* *contortus*	Albendazole (0.008–25 μg/mL)	EC_50_ *=* 3.55 mg/mL (EHA), 17.23 mg/mL (LDT)	Adamu et al., [76]
*# Calpurnia aurea*	Acetone	DBA	Leaves/flowers, stem	*Haemonchus* *contortus*	Albendazole (100% at 0.008–25 μg/mL)	EHA inhibition = 27% (leaves/flowers), 32% (stem)	Fouche et al., [75]
*Clausena anisata*	Acetone	DBA	Leaves	*Haemonchus* *contortus*	Albendazole (0.008–25 μg/mL)	EC_50_ *=* 1.8 mg/mL (EHA), 2.07 mg/mL (LDT)	Adamu et al., [76]
*Combretum apiculatum* Sond. Subsp. *Apiculatum*	Ethyl acetate, acetone, and water	DBA	Leaves	*Caenorhabditis* *elegans*	Levamisole	Ethyl acetate extract had 70–80% lethality at 1 mg/mL	McGaw et al., [80]
*Combretum bracteosum* (Hochst.) Engl. & Diels	Ethyl acetate, acetone, and water	DBA	Leaves	*Caenorhabditis* *elegans*	Levamisole	No activity observed	McGaw et al., [80]
*Combretum**celastroides* Welw ex Laws subsp. *Celastroides*	Ethyl acetate, acetone, and water	DBA	Leaves	*Caenorhabditis* *elegans*	Levamisole	No activity observed	McGaw et al., [80]
*Combretum**collinum* Fresen	Ethyl acetate, acetone, and water	DBA	Leaves	*Caenorhabditis* *elegans*	Levamisole	Acetone extract had 10–20% lethality at 0.5 and 1 mg/mL	McGaw et al., [80]
*Combretum edwardsii* Exell	Ethyl acetate, acetone, and water	DBA	Leaves	*Caenorhabditis* *elegans*	Levamisole	Acetone and ethyl acetate extracts had 10–20% lethality at 1 mg/mL	McGaw et al., [80]
*Combretum**erythrophyllum* (Burch.) Sond.	Ethyl acetate, acetone, and water	DBA	Leaves	*Caenorhabditis* *elegans*	Levamisole	Acetone and ethyl acetate extracts had 10–20% lethality at 1 mg/mL	McGaw et al., [80]
*Combretum**hereroense* Schinz	Ethyl acetate, acetone, and water	DBA	Leaves	*Caenorhabditis* *elegans*	Levamisole	Acetone extract had 20–30% lethality at 1 mg/mL	McGaw et al., [80]
*Combretum**imberbe* Wawra	Ethyl acetate, acetone, and water	DBA	Leaves	*Caenorhabditis* *elegans*	Levamisole	Acetone extract had 20–30% lethality at 1 mg/mL	McGaw et al., [80]
*Combretum**kraussii* Hochst. (Syn: *Combretum nelsonii*)	Ethyl acetate, acetone, and water	DBA	Leaves	*Caenorhabditis* *elegans*	Levamisole	Acetone extract had 20–30% lethality at 0.5 mg/mL	McGaw et al., [80]
*# Combretum microphyllum*	Ethyl acetate, acetone, and water	DBA	Leaves	*Caenorhabditis* *elegans*	Levamisole	Acetone and ethyl acetate extracts had 10–20% lethality at 0.5 and 1 mg/mL	McGaw et al., [80]
*Combretum mkuzense* J.D.Carr & Retief	Ethyl acetate, acetone, and water	DBA	Leaves	*Caenorhabditis* *elegans*	Levamisole	Acetone extract had 20–30% lethality at 1 mg/mL	McGaw et al., [80]
*Combretum moggi* Exell	Ethyl acetate, acetone, and water	DBA	Leaves	*Caenorhabditis* *elegans*	Levamisole	No activity observed	McGaw et al., [80]
*# Combretum molle* R.Br. ex. G.Don	Ethyl acetate, acetone, and water	DBA	Leaves	*Caenorhabditis* *elegans*	Levamisole	Acetone extract had 20–30% lethality at 1 mg/mL	McGaw et al., [80]
*Combretum mossambicense* (Klotzsch) Engl.	Ethyl acetate, acetone, and water	DBA	Leaves	*Caenorhabditis* *elegans*	Levamisole	Acetone extract had 20–30% lethality at 1 mg/mL	McGaw et al., [80]
*Combretum padoides* Engl.	Ethyl acetate, acetone, and water	DBA	Leaves	*Caenorhabditis* *elegans*	Levamisole	No activity observed	McGaw et al., [80]
*# Combretum paniculatum* Vent.	Ethyl acetate, acetone, and water	DBA	Leaves	*Caenorhabditis* *elegans*	Levamisole	Acetone extract had 10–20% lethality at 0.5 mg/mL	McGaw et al., [80]
*Combretum petrophilum* Retief	Ethyl acetate, acetone, and water	DBA	Leaves	*Caenorhabditis* *elegans*	Levamisole	Acetone extract had 10–20% lethality at 0.5 mg/mL	McGaw et al., [80]
*Combretum woodii*	Ethyl acetate, acetone, and water	DBA	Leaves	*Caenorhabditis* *elegans*	Levamisole	No activity observed	McGaw et al., [80]
*Combretum zeyheri*	Ethyl acetate, acetone, and water	DBA	Leaves	*Caenorhabditis* *elegans*	Levamisole	No activity observed	McGaw et al., [80]
*# Curtisia dentata*	Dichloromethane and acetone	DBA	Leaves	*Caenorhabditis* *elegans*, *Haemonchus contortus*, *Trichostrongylus colubriformis*	Levamisole (10 μg/mL)	Acetone extracts had the highest inhibition at 2.5 mg/mL after 2 h and 7 days of incubation. Both extracts inhibited the highest motility at 1.25–2.5 (*Haemonchus contortus*) and 0.63–2.5 mg/mL (*Trichostrongylus colubriformis*)	Shai et al., [81]
*# Cussonia spicata*	Hexane, methanol, and water	DBA	Roots	*Caenorhabditis* *elegans*	Levamisole (10 μg/mL)	No noteworthy activity	McGaw et al., [70]
*# Dombeya rotundifolia*	Hexane, methanol, and water	DBA	Aerial parts	*Caenorhabditis* *elegans*	Levamisole (10 μg/mL)	Water extract had 20% lethality at 1 and 2 mg/mL	McGaw et al., [70]
*# Euphorbia cupularis* (Syn: *Synadenium cupulare* (Boiss.)	Hexane, methanol, and water	DBA	Stem/leaves	*Caenorhabditis* *elegans*	Levamisole (10 μg/mL)	No noteworthy activity	McGaw et al., [70]
*Ficus sycomorus*	Acetone	DBA	Bark/stem, stem	*Haemonchus* *contortus*	Albendazole (100% at 0.008–25 μg/mL)	EHA inhibition = 25% (bark/stem), 21% (stem)	Fouche et al., [75]
*Heteromorpha trifoliata*	Acetone	DBA	Leaves	*Haemonchus* *contortus*	Albendazole (0.008–25 μg/mL)	EC_50_ *=* 0.62 mg/mL (EHA), 0.64 mg/mL (LDT)	Adamu et al., [76]
*# Hippobromus pauciflorus*	Hexane, methanol, and water	DBA	Aerial parts	*Caenorhabditis* *elegans*	Levamisole (10 μg/mL)	Hexane extract had 50% lethality at 2 mg/mL	McGaw et al., [70]
*Indigofera frutescens*	Acetone	DBA	Leaves	*Haemonchus* *contortus*	Albendazole (0.008–25 μg/mL)	EC_50_ *=* 7.11 mg/mL (EHA), 7.58 mg/mL (LDT)	Adamu et al., [76]
*Leucosidea sericea*	Acetone	DBA	Leaves	*Haemonchus* *contortus*	Albendazole (0.008–25 μg/mL)	EC_50_ *=* 1.08 mg/mL (EHA), 1.27 mg/mL (LDT)	Adamu et al., [76]
*Leucosidea sericea*	Petroleum ether, dichloromethane, ethanol, and water	CA	Leaves, stem	*Caenorhabditis* *elegans*	Levamisole (40 μg/mL)	Petroleum ether, dichloromethane, and ethanol leaf extracts had noteworthy anthelmintic effect (MLC = 0.26–0.52 mg/mL)	Aremu et al., [69]
*# Maerua angolensis*	Acetone	DBA	Stem, leaves	*Haemonchus* *contortus*	Albendazole (100% at 0.008–25 μg/mL)	EHA inhibition = 65% (stem), 25% (leaves)	Fouche et al., [75]
*Maesa lanceolata*	Acetone	DBA	Leaves	*Haemonchus* *contortus*	Albendazole (0.008–25 μg/mL)	EC_50_ *=* 0.72 mg/mL (EHA), 1.68 mg/mL (LDT)	Adamu et al., [76]
*Melia azedarach*	Acetone	DBA	Leaves	*Haemonchus* *contortus*	Albendazole (0.008–25 μg/mL)	EC_50_*=* 6.24 mg/mL (EHA), 10.96 mg/mL (LDT)	Adamu et al., [76]
*# Milletia grandis*	Acetone	DBA	Leaves	*Haemonchus* *contortus*	Albendazole (0.008–25 μg/m)	EC_50_*=* 5.57 mg/mL (EHA), 6.11 mg/mL (LDT)	Adamu et al., [76]
*# Schotia brachypetala* Sond	Hexane, methanol, and water	DBA	Leaves, bark	*Caenorhabditis* *elegans*	Levamisole (10 μg/mL)	All solvent extracts from, bark had 10% at 2 mg/mL. Hexane extract from leaves had 10% lethality at 2 mg/mL	McGaw et al., [70]
*# Sclerocarya birrea*	Acetone	DBA	Fruit	*Haemonchus* *contortus*	Albendazole (100% at 0.008–25 μg/mL)	EHA inhibition = 28%	Fouche et al., [75]
*# Sclerocarya birrea*	Hexane, methanol, and water	DBA	Bark	*Caenorhabditis* *elegans*	Levamisole (10 μg/mL)	Methanol extract had 40% lethality at 2 mg/mL	McGaw et al., [70]
*# Searsia lancea* (Syn: *Rhus lancea*)	Hexane, methanol, and water	DBA	Leaves, bark	*Caenorhabditis* *elegans*	Levamisole (10 μg/mL)	Hexane extracts had 50% (leaves) and 40% (bark) lethality at 2 mg/mL	McGaw et al., [70]
*# Senna petersiana*	Petroleum ether, dichloromethane, ethanol, and water	CA	Leaves	*Caenorhabditis* *elegans*	Levamisole (40 μg/mL)	Ethanol extract = 0.52 mg/mL	Aremu et al., [71]
*Strychnos mitis*	Acetone	DBA	Leaves	*Haemonchus* *contortus*	Albendazole (0.008–25 μg/mL)	EC_50_ *=* 16.56 mg/mL (EHA), 16.94 mg/mL (LDT)	Adamu et al., [76]
*# Tabernaemontana elegans*	Acetone	DBA	Leaves	*Haemonchus* *contortus*	Albendazole (100% at 0.008–25 μg/mL)	EHA inhibition = 47%	Fouche et al., [75]
*# Volkameria glabra* (*Clerodendrum glabrum*)	Acetone	DBA	Leaves	*Haemonchus* *contortus*	Albendazole (0.008–25 μg/mL)	EC_50_ *=* 1.48 mg/mL (EHA), 12.97 mg/mL (LDT)	Adamu et al., [76]
*# Zanthoxylum capense*	Acetone	DBA	Leaves	*Haemonchus* *contortus*	Albendazole (0.008–25 μg/mL)	EC_50_ *=* 13.26 mg/mL (EHA), 13.64 mg/mL (LDT)	Adamu et al., [76]
*# Ziziphus mucronata*	Hexane, methanol, and water	DBA	Bark, leaves	*Caenorhabditis* *elegans*	Levamisole (10 μg/mL)	No noteworthy activity	McGaw et al., [70]

**Table 5 vetsci-08-00228-t005:** Examples of in vitro antioxidant effect of woody plants used for ethnoveterinary medicine in South Africa. # Plant species: denotes woody plants with ethnoveterinary uses in Table 2. ABTS—2,2′-azinobis-(3-ethylbenzothiazoline-6-sulfonic acid), DPPH—1,1-diphenyl-2-picryl-hydrazyl, FRAP—ferric reducing antioxidant power, TEAC—trolox equivalent antioxidant assay.

# Plant Species	Assay Type	Plant Part	Findings	Reference
*Alsophila dregei* (Kunze) R.M.Tryon (Syn: *Cyathea dregei*)	DPPH	Leaves	EC_50_ = 3 µg/mL	Adamu et al., [68]
*Alsophila dregei* (Kunze) R.M.Tryon (Syn: *Cyathea dregei*)	ABTS	Leaves	0.4 TEAC	Adamu et al., [68]
*Apodytes dimidiata* E.Mey. ex. Arn.	DPPH	Leaves	EC_50_ = 3.5 µg/mL	Adamu et al., [68]
*Apodytes dimidiata* E.Mey. ex. Arn.	ABTS	Leaves	0.3 TEAC	Adamu et al., [68]
*Brachylaena discolor* DC.	DPPH	Leaves	EC_50_ = 2.6 µg/mL	Adamu et al., [68]
*Brachylaena discolor* DC.	ABTS	Leaves	0.2 TEAC	Adamu et al., [68]
*Burkea africana* Hook.	DPPH	Leaves	IC_50_ = 3.55 µg/mL	Dzoyem and Eloff [85]
*Burkea africana* Hook.	ABTS	Leaves	IC_50_ = 3.21 µg/mL	Dzoyem and Eloff [85]
*Burkea africana* Hook.	FRAP	Leaves	IC_50_ = 231.07 µg Fe (II)/g	Dzoyem and Eloff [85]
*Clausena anisata* (Willd.) Hook.f. ex. Benth.	DPPH	Leaves	EC_50_ = 2.5 µg/mL	Adamu et al., [68]
*Clausena anisata* (Willd.) Hook.f. ex. Benth.	ABTS	Leaves	0.2 TEAC	Adamu et al., [68]
*Combretum zeyheri* Sond.	DPPH	Leaves	IC_50_ = 3.52 µg/mL	Dzoyem and Eloff [85]
*Combretum zeyheri* Sond.	ABTS	Leaves	IC_50_ = 4.64 µg/mL	Dzoyem and Eloff [85]
*Combretum zeyheri* Sond.	FRAP	Leaves	IC_50_ = 95.98 µg Fe (II)/g	Dzoyem and Eloff [85]
*Dalbergia nitidula* Welw. ex. Baker	DPPH	Leaves	IC_50_ = 9.31 μg/mL	Dzoyem et al., [67]
*Dalbergia nitidula* Welw. ex. Baker	ABTS	Leaves	IC_50_ = 21.3 μg/mL	Dzoyem et al., [67]
*# Englerophytum magalismontanum* (Sond.) T.D.Penn	DPPH	Leaves	IC_50_ = 10.8 µg/mL	Dzoyem and Eloff [85]
*# Englerophytum magalismontanum* (Sond.) T.D.Penn	ABTS	Leaves	IC_50_ = 12.22 µg/mL	Dzoyem and Eloff [85]
*# Englerophytum magalismontanum* (Sond.) T.D.Penn	FRAP	Leaves	IC_50_ = 76 µg Fe (II)/g	Dzoyem and Eloff [85]
*#**Erythrina caffra* Thunb.	DPPH	Leaves	IC_50_ = 268.6 μg/mL	Dzoyem et al., [67]
*#**Erythrina caffra* Thunb.	ABTS	Leaves	IC_50_ = 173.28 μg/mL	Dzoyem et al., [67]
*Euclea undulata* Thunb.	DPPH	Leaves	31.66 µg/mL	Dzoyem and Eloff [85]
*Euclea undulata* Thunb.	ABTS	Leaves	32.67 µg/mL	Dzoyem and Eloff [85]
*Euclea undulata* Thunb.	FRAP	Leaves	274.19 µg Fe (II)/g	Dzoyem and Eloff [85]
*Heteromorpha trifoliata* (H.L.Wendl.) Eckl. & Zeyh.	DPPH	Leaves	EC_50_ = 4.36 µg/mL	Adamu et al., [68]
*Heteromorpha trifoliata* (H.L.Wendl.) Eckl. & Zeyh.	ABTS	Leaves	0.2 TEAC	Adamu et al., [68]
*Indigofera frutescens* L.f.	DPPH	Leaves	EC_50_ = 0 µg/mL	Adamu et al., [68]
*Indigofera frutescens* L.f.	ABTS	Leaves	0.5 TEAC	Adamu et al., [68]
*Indigofera frutescens* L.f.	DPPH	Leaves	IC_50_ = 22.31 μg/mL	Dzoyem et al., [67]
*Indigofera frutescens* L.f.	ABTS	Leaves	IC_50_ = 134.64 μg/mL	Dzoyem et al., [67]
*# Jatropha curcas* L.	DPPH	Leaves	IC_50_ = 137.08 µg/mL	Dzoyem and Eloff [85]
*# Jatropha curcas* L.	ABTS	Leaves	IC_50_ = 115.23 µg/mL	Dzoyem and Eloff [85]
*# Jatropha curcas* L.	FRAP	Leaves	IC_50_ = 68.17 µg Fe (II)/g	Dzoyem and Eloff [85]
*Leucaena leucocephala* (Lam.) de Wit	DPPH	Leaves	IC_50_ = 9.86 µg/mL	Dzoyem and Eloff [85]
*Leucaena leucocephala* (Lam.) de Wit	ABTS	Leaves	IC_50_ = 9.85 µg/mL	Dzoyem and Eloff [85]
*Leucaena leucocephala* (Lam.) de Wit	FRAP	Leaves	IC_50_ = 289.27 µg Fe (II)/g	Dzoyem and Eloff [85]
*Leucosidea sericea*	DPPH	Leaves	EC_50_ = 0.0 µg/mL	Adamu et al., [68]
*Leucosidea sericea*	ABTS	Leaves	0.7 TEAC	Adamu et al., [68]
*Maesa lanceolata* Forssk.	DPPH	Leaves	EC_50_ = 1.4 µg/mL	Adamu et al., [68]
*Maesa lanceolata* Forssk.	ABTS	Leaves	1.2 TEAC	Adamu et al., [68]
*Melia azedarach* L.	DPPH	Leaves	EC_50_ = 3.3 µg/mL	Adamu et al., [68]
*Melia azedarach* L.	ABTS	Leaves	0.8 TEAC	Adamu et al., [68]
*# Millettia grandis* (E.Mey.) Skeels	DPPH	Leaves	EC_50_ = 4.6 µg/mL	Adamu et al., [68]
*# Millettia grandis* (E.Mey.) Skeels	ABTS	Leaves	0.6 TEAC	Adamu et al., [68]
*Morus mesozygia* Stapf	DPPH	Leaves	IC_50_ = 15.85 µg/mL	Dzoyem and Eloff [85]
*Morus mesozygia* Stapf	ABTS	Leaves	IC_50_ = 271.86 µg/mL	Dzoyem and Eloff [85]
*Morus mesozygia* Stapf	FRAP	Leaves	IC_50_ = 127.34 µg Fe (II)/g	Dzoyem and Eloff [85]
*Philenoptera nelsii* (Schinz) Schrire (*Lonchocarpus nelsii*)	DPPH	Leaves	IC_50_ = 247.7 μg/mL	Adamu et al., [68]
*Philenoptera nelsii* (Schinz) Schrire (*Lonchocarpus nelsii*)	ABTS	Leaves	IC_50_ = 41.39 μg/mL	Adamu et al., [68]
*Strychnos mitis* S.Moore	DPPH	Leaves	EC_50_ = 3.5 µg/mL	Adamu et al., [68]
*Strychnos mitis* S.Moore	ABTS	Leaves	0.3 TEAC	Adamu et al., [68]
*Uapaca nitida* Müll.Arg.	DPPH	Leaves	IC_50_ = 125.86 µg/mL	Dzoyem and Eloff [85]
*Uapaca nitida* Müll.Arg.	ABTS	Leaves	IC_50_ = 28.81 µg/mL	Dzoyem and Eloff [85]
*Uapaca nitida* Müll.Arg.	FRAP	Leaves	IC_50_ = 177.32 µg Fe (II)/g	Dzoyem and Eloff [85]
*# Volkameria glabra* (E. Mey.) Mabb. & Y. W. Yuan (Syn: *Clerodendrum* *glabrum*)	DPPH	Leaves	EC_50_ = 3.5 µg/mL	Adamu et al., [68]
*# Volkameria glabra* (E. Mey.) Mabb. & Y. W. Yuan (Syn: *Clerodendrum* *glabrum*)	ABTS	Leaves	0.5 TEAC	Adamu et al., [68]
*# Zanthoxylum capense* (Thunb.) Harv.	DPPH	Leaves	EC_50_ = 4 µg/mL	Adamu et al., [68]
*# Zanthoxylum capense* (Thunb.) Harv.	ABTS	Leaves	0.4 TEAC	Adamu et al., [68]

**Table 6 vetsci-08-00228-t006:** Cytotoxic activity of woody plants used for ethnoveterinary purposes in South Africa. # Plant species: denotes woody plants with ethnoveterinary uses in Table 2; * Test system: MTT—3–5-dimethyl thiazol-2-yl-2, 5-diphenyl tetrazolium bromide.

# Plant Species	Solvent	Plant Part	* Test System	Positive Control	Findings	Reference
*Alsophila dregei* (Syn: *Cyathea dregei*)	Acetone	Leaves	Tetrazolium-based colorimetric MTT assay using Vero monkey kidney cells	Berberine chloride	LC_50_ = 0.00332 mg/mL	Adamu et al., [76]
*Apodytes dimidiata*	Acetone	Leaves	Tetrazolium-based colorimetric MTT assay using Vero monkey kidney cells	Berberine chloride	LC_50_ = 0.00396 mg/mL	Adamu et al., [76]
*Berchemia zeyheri*	Hexane, methanol, and water	Bark	Brine shrimp lethality/toxicity using *Artemia salina*	Podophyllotoxin (7 μg/mL)	Water extract had the highest lethal effect (LC_50_ = 3.9 mg/mL)	McGaw et al., [70]
# *Bolusanthus speciosus*	Acetone	Leaves	Tetrazolium-based colorimetric MTT assay using Vero monkey kidney cells	Doxorubicin = 1.76 µg/mL	LC_50_ = 52.8 μg/mL	Elisha et al., [66]
*Brachylaena discolor*	Acetone	Leaves	Tetrazolium-based colorimetric MTT assay using Vero monkey kidney cells	Berberine chloride	LC_50_ = 0.00752 mg/mL	Adamu et al., [76]
# *Calpurnia aurea*	Acetone	Leaves	Tetrazolium-based colorimetric MTT assay using Vero monkey kidney cells	Doxorubicin = 1.76 µg/mL	LC_50_ = 13.6 μg/mL	Elisha et al., [66]
# *Calpurnia aurea*	Acetone	Leaves/flowers, stem	Tetrazolium-based colorimetric MTT assay using Vero monkey kidney cells	Doxorubicin (2.97 μg/mL)	Leaves/flowers, LC_50_ = 166.63 μg/mL, Stem LC_50_ = 223.97 μg/mL	Fouche et al., [75]
*Clausena anisata*	Acetone	Leaves	Tetrazolium-based colorimetric MTT assay using Vero monkey kidney cells	Berberine chloride	LC_50_ = 0.17186 mg/mL	Adamu et al., [76]
*Cremaspora triflora*	Acetone	Leaves	Tetrazolium-based colorimetric MTT assay using Vero monkey kidney cells	Doxorubicin = 1.76 µg/mL	LC_50_ = 57.4 μg/mL	Elisha et al., [66]
# *Cussonia spicata*	Hexane, methanol, and water	Roots	Brine shrimp lethality/toxicity using *Artemia salina*	Podophyllotoxin (7 μg/mL)	Water extract had the highest lethal effect (LC_50_ = 2.6 mg/mL)	McGaw et al., [70]
*Dalbergia nitidula*	Acetone	Leaves	Tetrazolium-based colorimetric MTT assay using Vero monkey kidney cells	Doxorubicin (2.29 μg/mL)	LC_50_ = 51.28 μg/mL	Dzoyem et al., [67]
# *Dombeya rotundifolia*	Hexane, methanol, and water	Aerial part	Brine shrimp lethality/toxicity using *Artemia salina*	Podophyllotoxin (7 μg/mL)	All extracts had no lethal effect	McGaw et al., [70]
*Elaeodendron croceum* (Thunb.) DC.	Acetone	Leaves	Tetrazolium-based colorimetric MTT assay using Vero monkey kidney cells	Doxorubicin = 1.76 µg/mL	LC_50_ = 5.2 μg/mL	Elisha et al., [66]
# *Erythrina caffra*	Acetone	Leaves	Tetrazolium-based colorimetric MTT assay using Vero monkey kidney cells	Doxorubicin (2.29 μg/mL)	LC_50_ = 19.93 μg/mL	Dzoyem et al., [67]
# * Euphorbia cupularis* (Syn: *Synadenium cupulare*)	Hexane, methanol, and water	Aerial part	Brine shrimp lethality/toxicity using *Artemia salina*	Podophyllotoxin (7 μg/mL)	All extracts had no lethal effect	McGaw et al., [70]
*Ficus sycomorus*	Acetone	Bark/stem, stem	Tetrazolium-based colorimetric MTT assay using Vero monkey kidney cells	Doxorubicin (2.97 μg/mL)	LC_50_ = 172.94 μg/mL (bark/stem), LC_50_ = 48.74 μg/mL (stem)	Fouche et al., [75]
# *Heteromorpha arborescens*	Acetone	Leaves	Tetrazolium-based colorimetric MTT assay using Vero monkey kidney cells	Doxorubicin = 1.76 µg/mL	LC_50_ = 81.0 μg/mL	Elisha et al., [66]
*Heteromorpha trifoliata*	Acetone	Leaves	Tetrazolium-based colorimetric MTT assay using Vero monkey kidney cells	Berberine chloride	LC_50_ = 0.04252 mg/mL	Adamu et al., [76]
# *Hippobromus pauciflorus*	Hexane, methanol, and water	Aerial part	Brine shrimp lethality/toxicity using *Artemia salina*	Podophyllotoxin (7 μg/mL)	All extract had no lethal effect	McGaw et al., [70]
*Indigofera cylindrica*	Acetone	Leaves	Tetrazolium-based colorimetric MTT assay using Vero monkey kidney cells	Doxorubicin (2.29 μg/mL)	LC_50_ = 77.59 μg/mL	Dzoyem et al., [67]
*Indigofera frutescens*	Acetone	Leaves	Tetrazolium-based colorimetric MTT assay using Vero monkey kidney cells	Berberine chloride	LC_50_ = 0.1044 mg/mL	Adamu et al., [76]
*Leucosidea sericea*	Acetone	Leaves	Tetrazolium-based colorimetric MTT assay using Vero monkey kidney cells	Berberine chloride	LC_50_ = 0.0515 mg/mL	Adamu et al., [76]
*Lonchocarpus nelsii*	Acetone	Leaves	Tetrazolium-based colorimetric MTT assay using Vero monkey kidney cells	Doxorubicin (2.29 μg/mL)	LC_50_ = 81.09 μg/mL	Dzoyem et al., [67]
# *Maerua angolensis*	Acetone	Stem, leaves	Tetrazolium-based colorimetric MTT assay using Vero monkey kidney cells	Doxorubicin (2.97 μg/mL)	LC_50_ = 180.64 μg/mL (stem), LC_50_ = 73.76 μg/mL (leaves)	Fouche et al., [75]
*Maesa lanceolata*	Acetone	Leaves	Tetrazolium-based colorimetric MTT assay using Vero monkey kidney cells	Berberine chloride	LC_50_ = 0.01577 mg/mL	Adamu et al., [76]
*Maesa lanceolata*	Acetone	Leaves	Tetrazolium-based colorimetric MTT assay using Vero monkey kidney cells	Doxorubicin = 1.76 µg/mL	LC_50_ = 0.38 μg/mL	Elisha et al., [66]
*Melia azedarach*	Acetone	Leaves	Tetrazolium-based colorimetric MTT assay using Vero monkey kidney cells	Berberine chloride	LC_50_ = 0.14466 mg/mL	Adamu et al., [76]
# *Milletia grandis*	Acetone	Leaves	Tetrazolium-based colorimetric MTT assay using Vero monkey kidney cells	Berberine chloride	LC_50_ = 0.05336 mg/mL	Adamu et al., [76]
*Morus mesozygia*	Acetone	Leaves	Tetrazolium-based colorimetric MTT assay using Vero monkey kidney cells	Doxorubicin = 1.76 µg/mL	LC_50_ = 40.7 μg/mL	Elisha et al., [66]
# *Pittosporum viridiflorum*	Acetone	Leaves	Tetrazolium-based colorimetric MTT assay using Vero monkey kidney cells	Doxorubicin = 1.76 µg/mL	LC_50_ = 54.6 μg/mL	Elisha et al., [66]
*# Pterocarpus angolensis*	Hexane, methanol, and water	Bark, leaves	Brine shrimp lethality/toxicity using *Artemia salina*	Podophyllotoxin (7 μg/mL)	All extracts from the bark had no lethal effect. Hexane and methanol extracts from the leaves had the highest lethal effect (LC_50_ = 3.6–3.8 mg/mL)	McGaw et al., [70]
# *Sclerocarya birrea*	Acetone	Fruit	Tetrazolium-based colorimetric MTT assay using Vero monkey kidney cells	Doxorubicin (2.97 μg/mL)	LC_50_ = 214.79 μg/mL	Fouche et al., [75]
# *Sclerocarya birrea*	Hexane, methanol, and water	Bark	Brine shrimp lethality/toxicity using *Artemia salina*	Podophyllotoxin (7 μg/mL)	All extracts had no lethal effect	McGaw et al., [70]
# *Searsia lancea* (*Rhus lancea*)	Hexane, methanol, and water	Bark, leaves	Brine shrimp lethality/toxicity using *Artemia salina*	Podophyllotoxin (7 μg/mL)	Water extract from the bark (LC_50_ = 3.9 mg/mL) and leaves (LC_50_ = 0.6 mg/mL) had the highest toxic effect	McGaw et al., [70]
*Strychnos mitis*	Acetone	Leaves	Tetrazolium-based colorimetric MTT assay using Vero monkey kidney cells	Berberine chloride	LC_50_ = 0.01721 mg/mL	Adamu et al., [76]
# *Tabernaemontana elegans*	Acetone	Leaves	Tetrazolium-based colorimetric MTT assay using Vero monkey kidney cells	Doxorubicin (2.97 μg/mL)	LC_50_ = 32.35 μg/mL	Fouche et al., [75]
# *Tetradenia riparia*	Acetone and water	Flowers, leaves	Tetrazolium-based colorimetric MTT assay using Vero monkey kidney cells	Doxorubicin (5.4326 µM)	LC_50_, flowers = 0.0823 mg/mL (acetone), 0.1784 mg/mL (water); leaves = 0.0513 mg/mL (acetone), 0.2738 mg/mL (water)	Sserunkuma et al., [86]
# *Vachellia nilotica* (*Acacia nilotica*)	Acetone and water	Bark, leaves	Tetrazolium-based colorimetric MTT assay using Vero monkey kidney cells	Doxorubicin (5.4326 µM)	LC_50_, bark = 0.0332 mg/mL (acetone), 0.0278 mg/mL (water); leaves = 0.2187 mg/mL (acetone), 0.0688 mg/mL (water)	Sserunkuma et al., [86]
*Virgilia divaricata* Adamson	Acetone	Leaves	Tetrazolium-based colorimetric MTT assay using Vero monkey kidney cells	Doxorubicin (2.29 μg/mL)	LC_50_ = 30.08 μg/mL	Dzoyem et al., [67]
# * Volkameria glabra* (*Clerodendrum glabrum*)	Acetone	Leaves	Tetrazolium-based colorimetric MTT assay using Vero monkey kidney cells	Berberine chloride	LC_50_ = 0.04251 mg/mL	Adamu et al., [76]
# *Zanthoxylum capense*	Acetone	Leaves	Tetrazolium-based colorimetric MTT assay using Vero monkey kidney cells	Berberine chloride	LC_50_ = 0.02095 mg/mL	Adamu et al., [76]
# *Ziziphus mucronata*	Hexane, methanol and water	Bark, leaves	Brine shrimp lethality/toxicity using *Artemia salina*	Podophyllotoxin (7 μg/mL)	Bark extracts had no lethal effect. Hexane extract of leaves had LC_50_ = 0.9 mg/mL	McGaw et al., [70]

**Table 7 vetsci-08-00228-t007:** Phytochemical analysis (based on spectrophotometric method) of woody plants used in ethnoveterinary medicine. # Plant species: denotes woody plants with ethnoveterinary uses in Table 2; GAE—gallic acid equivalents, TPC—total phenolic content, TFC—total flavonoid content, CT—condensed tannin, GC—gallotannin content, LCE—leucocyanidin equivalents, CTE—catechin equivalents, QE—quercetin equivalents.

# Plant Species	Plant Part	Extract	Findings	Reference
# *Acokanthera oppositifolia*	Leaves, twigs	50% Methanol	TPC = 2.5 and 7.2 mg GAE/g GC = 2 µg and 5.2 µg GAE/g CT = 0.005% and 0.12% LCE/g TFC = 0.002 and 0.001 mg CTE/g	Aremu et al., [71]
*Burkea africana*	Leaves	Acetone	TPC = 14.39 mg GAE/g	Dzoyem and Eloff [85]
*Combretum zeyheri*	Leaves	Acetone	TPC = 3.29 mg GAE/g	Dzoyem and Eloff [85]
# *Curtisia dentata*	Stem bark	Acetone	TPC = 8.94 mg GAE/g	Olaokun et al., [90]
*Dalbergia nitidula*	Leaves	Acetone	TPC = 1.51 mg GAE/g	Dzoyem et al., [67]
# *Englerophytum magalismontanum*	Leaves	Acetone	TPC = 0.86 mg GAE/g	Dzoyem and Eloff [85]
# *Erythrina caffra*	Leaves	Acetone	TPC = 150.82 mg/g GAE TFC = 72.8 mg QE/g	Dzoyem et al., [67]
*Euclea undulata*	Leaves	Acetone	TPC = 234.56 mg/g GAE TFC = 64.36 mg QE/g	Dzoyem and Eloff [85]
*Indigofera frutescens* (*Indigofera cylindrical*)	Leaves	Acetone	TPC = 125.12 mg/g GAE TFC = 27.69 mg/g QE	Dzoyem et al., [67]
# *Jatropha curcas*	Leaves	Acetone	TPC = 100.89 mg/g GAE TFC = 68.43 mg QE/g	Dzoyem and Eloff [85]
*Leucaena leucocephala*	Leaves	Acetone	TPC = 129.78 mg/g GAE TFC = 35.16 mg QE/g	Dzoyem and Eloff [85]
*Leucosidea sericea*	Leaves, stem	50% Methanol	TPC = 36.66 and 6.4 mg GAE/g GC = 29.32 and 5.12 µg GAE/g CT = 0.46 and 0.47% LCE/g TFC = 0.66 and 0.26 mg CTE/g	Aremu et al., [69]
*Lippia javanica* (Burm.f) Spreng	Leaves	Acetone	TPC = 130.12 mg GAE/g	Dzoyem and Eloff [85]
*Morus mesozygia*	Leaves	Acetone	TPC = 427.53 mg/g GAE TFC = 80.72 mg QE/g	Dzoyem and Eloff [85]
*Philenoptera nelsii* (*Lonchocarpus nelsii*)	Leaves	Acetone	TPC = 258.4 mg/g GAE TFC = 159.61 mg QE/g	Dzoyem et al., [67]
# * Pittosporum viridiflorum* Sims	Stem bark	Acetone	TPC = 181.49 mg/g GAE TFC = 13.75 mg QE/g	Olaokun et al., [90]
*# Senna petersiana*	Leaves	50% Methanol	TPC = 5 mg GAE/g GC = 4 µg GAE/g CT = 0.18% LCE/g TFC = 0.1 mg CTE/g	Aremu et al., [71]
*Uapaca nitida*	Leaves	Acetone	TPC = 26.08 mg/g GAE TFC = 20.31 mg/g QE	Dzoyem and Eloff [85]
*Virgilia divaricata* Adamson	Leaves	Acetone	TPC = 137.3 mg/g GAE TFC = 15.3 mg QE/g	Dzoyem et al., [67]
*Ziziphus rivularis* Codd	Leaves	Acetone	TPC = 182.79 mg/g GAE TFC = 46.88 mg QE/g	Dzoyem and Eloff [85]

## Data Availability

All data are included as part of the manuscript.

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
