# Peer review of "A Review of Ethnoveterinary Knowledge, Biological Activities and Secondary Metabolites of Medicinal Woody Plants Used for Managing Animal Health in South Africa"

_vetsci, 2021, doi:10.3390/vetsci8100228_

Round 1

Reviewer 1 Report

The manuscript presents an extensive description of the ethnovet use of plants in South Africa. The inclusion and exclusion criteria are supported by the objectives of the study. The data presentation format is clear. In order to promote knowledge of the vegetables used in ethnoveterinary medicine in Africa, I suggest that the manuscript be published in this format.

Author Response

Dear Reviewer,

Thank you for the positive consideration and valuable time

Reviewer 2 Report

This is a well prepared and an interesting review.

Authors may add some information about oregano or carvacrol rich compounds.

Author Response

Dear Reviewer,

Thank you for the suggestions to highlight oregano or carvacrol rich compounds however, this aspect is not aligned to the inclusion and exclusion criteria. We also do not have the presence of these compounds in the plants included in the review article

Reviewer 3 Report

The authors present a very interesting extensive review of ethnoveterinary knowledge in South Africa. The study can be published after a few corrections.

Advice: some plants have had recent changes in classification. Please review the scientific name of the plants cited.

It is also important that the discussion that the collection of some therapeutic properties from plants should not lead to their indiscriminate use, the replacement of treatments with scientific evidence and the potential toxicity of some plants (and plant ecotypes; often variable from region to region ) when consumed under certain circumstances.

Line 17: "Globally, the increasing importance of ethnoveterinary medicine as remedies for animal health have been recognised..." --> Rephrase the sentence. From a scientific point of view, the potential of knowledge should not be underestimated. But it must always be stressed that its scientific validation is always necessary, as the authors emphasize further in the text.

Line 189: please remove "to overcome therapeutic failure" --> The previous sentence is enough to emphasize or that it is intended. To add that it is just to avoid therapeutic failures, and not referring to the problems of public health from a prespective One Health, it can induce an incomplete prespective

Author Response

The authors present a very interesting extensive review of ethnoveterinary knowledge in South Africa. The study can be published after a few corrections.

Advice: some plants have had recent changes in classification. Please review the scientific name of the plants cited.

RESPONSE: As recommended, we have verified the names of all the plants using ‘The World Flora Online’ (http://www.worldfloraonline.org/). We fully acknowledge the importance of correct scientific names from Ethnobotany and Ethnopharmacology perspectives (Rivera et al 2014; Weckerle et al 2018). Please see Table 2 for the revised names highlighted in Red font

Rivera, D., Allkin, R., Obón, C., Alcaraz, F., Verpoorte, R., Heinrich, M. 2014. What is in a name? The need for accurate scientific nomenclature for plants. Journal of Ethnopharmacology 152:393-402.

Weckerle, C.S., de Boer, H.J., Puri, R.K., van Andel, T., Bussmann, R.W., Leonti, M. 2018. Recommended standards for conducting and reporting ethnopharmacological field studies. Journal of Ethnopharmacology 210:125-132.

It is also important that the discussion that the collection of some therapeutic properties from plants should not lead to their indiscriminate use, the replacement of treatments with scientific evidence and the potential toxicity of some plants (and plant ecotypes; often variable from region to region ) when consumed under certain circumstances.

RESPONSE: Some of these aforementioned salient points have been briefly highlighted in the conclusion of the revised manuscript.

Line 17: "Globally, the increasing importance of ethnoveterinary medicine as remedies for animal health have been recognised..." --> Rephrase the sentence. From a scientific point of view, the potential of knowledge should not be underestimated. But it must always be stressed that its scientific validation is always necessary, as the authors emphasize further in the text.

RESPONSE: We have rephrased the sentence to read ‘Globally, the use of ethnoveterinary medicine as remedies for animal health among different eth-nic groups justify the need for a systematic exploration to enhance their potential.’

Line 189: please remove "to overcome therapeutic failure" --> The previous sentence is enough to emphasize or that it is intended. To add that it is just to avoid therapeutic failures, and not referring to the problems of public health from a prespective One Health, it can induce an incomplete prespective

RESPONSE: We have removed ‘to overcome therapeutic failure’ as recommended (please see section 3.4, first sentence)